# Oocyte-derived microvilli control female fertility by optimizing ovarian follicle selection in mice

Yan Zhang[1,4], Ye Wang[1,4], Xie'an Feng[1], Shuo Zhang[1], Xueqiang Xu[1], Lingyu Li[1], Shudong Niu[1], Yingnan Bo[1], Chao Wang[1], Zhen Li[2], Guoliang Xia[1,3] & Hua Zhang [1✉]

Crosstalk between oocytes and surrounding somatic cells is crucial for mammalian oogenesis, but the structural mechanisms on oocytes to control female reproduction remain unknown. Here we combine endogenous-fluorescent tracing mouse models with a high-resolution live-cell imaging system to characterize oocyte-derived mushroom-like microvilli (Oo-Mvi), which mediate germ-somatic communication in mice. We perform 3D live-cell imaging to show that Oo-Mvi exhibit cellular characteristics that fit an exocrine function for signaling communication. We find that deletion of the microvilli-forming gene *Radixin* in oocytes leads to the loss of Oo-Mvi in ovaries, and causes a series of abnormalities in ovarian development, resulting in shortened reproductive lifespan in females. Mechanistically, we find that Oo-Mvi enrich oocyte-secreted factors and control their release, resulting in optimal selection of ovarian follicles. Taken together, our data show that the Oo-Mvi system controls the female reproductive lifespan by governing the fate of follicles.

[1] State Key Laboratory of Agrobiotechnology, College of Biological Sciences, China Agricultural University, Beijing, China. [2] State Key Laboratory of Plant Physiology and Biochemistry, College of Biological Sciences, China Agricultural University, Beijing, China. [3] Key Laboratory of Ministry of Education for Conservation and Utilization of Special Biological Resources in Western China, College of Life Science, Ningxia University, Yinchuan, Ningxia, China. [4] These authors contributed equally: Yan Zhang, Ye Wang. ✉email: huazhang@cau.edu.cn

   1

In animals with sexual reproduction, female oocytes are primarily responsible for supporting the early development of their offspring. Therefore, a robust and finely regulated development of oocytes in adult life is the foundation of species continuity[1,2]. In mammals, the oocyte is covered by surrounding somatic cells (granulosa cells, GCs) to form the ovarian follicle as the functional unit of female reproduction[3]. The synchronous development of oocyte and GCs orchestrates programmed folliculogenesis to guarantee the production of qualified egg[2,4]. However, growing oocytes establish a glycoprotein egg coat, the zona pellucida (ZP), blocking the direct cellular membrane attachment between oocytes and GCs[5,6].

To overcome the blockage of the ZP, microvilli-like structures existing in the ZP are believed to construct communications between oocyte and GCs, maintaining proper folliculogenesis[7]. Previous results from electron microscopy and cytoskeleton immunostaining studies showed that those structures, named transzonal projections (TZPs), are derived from the GCs of follicles in mammals[8–10]. Several reports have also suggested that oocytes might form microvillus structures to participate in follicle development[11,12]. The lack of direct in vivo experimental evidence, however, made the cellular origin of those communicating structures in the mammalian ovary elusive[13–15]. Meanwhile, the inner mechanisms regulating the establishment of those structures, and the physiological significance of those microvilli in female reproduction have not been well identified.

In the mammalian ovary, a complicated but orchestrated regulating system determines the developmental fate of each individual follicle, and the balance of follicle survival and death determines the proper female reproductive lifespan[3,16]. Inside the follicles, the pivotal role of oocytes governing follicle development has been well accepted[2,4]. By releasing various specific oocyte-secreted factors (OSFs), such as growth differentiation factor 9 (GDF9) and bone morphogenic protein 15 (BMP15)[17–19], oocytes control the proliferation and activity of GCs to determine the fate of follicles[2]. However, as soluble factors, OSFs need to pass through the ZP to activate downstream signaling in GCs, but the existence of the ZP increases the physical distance between oocytes and GCs, which might reduce the efficiency of OSFs. The cellular structural basis of the oocyte to efficiently release the OSFs is not clear, and the mechanisms of how oocyte overcomes the ZP blockage to stimulate GC activity is unknown.

In the current study, we report an oocyte-derived microvilli system that mediates orderly and timely OSF release, which contributes to the fine regulation of ovarian follicle development and female reproductive lifespan maintenance. By combining endogenous-fluorescent tracing mouse models with a high-resolution live-cell imaging system, we establish systems to image the developmental details of mouse ovarian follicle cells at the single-cell and subcellular levels. With this system, we trace and describe the derivation and cellular behaviors of microvilli structures in ovarian follicles in detail. Our imaging observations show a cellular character of oocyte microvilli (Oo-Mvi), which play regulating roles in the development of ovarian follicles. By deleting *Radixin* (*Rdx*), which is a gene belonging to the microvillus-forming related ERM (*Ezrin/Radixin/Moesin*) gene family in oocytes[20,21], we disrupt the formation of Oo-Mvi and determine how these structures regulate the release of OSFs to govern female reproduction in mice. With these experimental data, we identify an elegant system by which Oo-Mvi in the ZP determine the fate of individual follicles by regulating oocyte-GC communication, which in turn adjusts the balance of follicle survival and death to determine the reproductive lifespan in female mammals.

## Results

### Imaging microvilli structures on oocyte by endogenous-fluorescent mouse models.
As a conserved communicating structure in the ZP, microvilli or TZPs have been well documented to exist between oocytes and GCs[22]. To visualize these structures and distinguish their cellular origin during follicle development, an *mTmG* reporter mouse model in which membrane-localized GFP (mG) is inducibly expressed in Cre-positive cells[23], was introduced in this study. As illustrated in Fig. 1a, by crossing the *mTmG* mice with oocyte-specific *Gdf9-Cre* mice[24] or granulosa cell-specific *Foxl2-CreER^{T2}* mice[25], we established a system to distinguish the derivation of microvilli by drawing the outlines of the cellular membrane with mG.

We first examined the ovaries in tamoxifen-treated *Foxl2-CreER^{T2};mTmG* females (Supplementary Fig. 1a), intensive GC-TZPs were observed in the ZP of growing follicles (Fig. 1b, arrows) under spinning-disc confocal microscopy. By modifying tamoxifen dosage, which allowed a few of GCs to be labeled and imaged, we reconstructed the 3D model of GC membrane and revealed a messy tree root-like structure of GC-TZPs at the single-cell level (Fig. 1b, right). These results are consistent with previous observations of GC-TZPs[10] and further demonstrated that GC-TZPs play a supporting function in oocyte development[7].

Interestingly, in *Gdf9-Cre;mTmG* ovaries (Supplementary Fig. 1b), a remarkably low density of microvilli structures derived from oocytes were found in the ZP in growing follicles (Fig. 1c, arrowheads). These Oo-Mvi exhibited a mushroom-like structure that consisted of a slender handle and a swollen vesicle tip (Fig. 1c, right). Histologically, we found that the establishment of Oo-Mvi was consistent with the growth of follicles (Fig. 1d). With the formation of the ZP, Oo-Mvi were first observed on the membrane of oocytes in primary follicles (Fig. 1d, arrowheads) and existed in the ZP at all growing stages (Fig. 1d, arrowheads), showing that the function of Oo-Mvi should be related to the growth of ovarian follicles. We next investigated the behaviors of Oo-Mvi in living oocytes isolated from adult *Gdf9-Cre;mTmG* ovaries. As shown in Fig. 1e, Oo-Mvi were clearly observed on the surface of oocytes under a live-cell imaging system (Supplementary Movie 1). Over the course of 50 min, no remarkable growth or extension of Oo-Mvi was observed (Supplementary Movie 2), showing that the formation of Oo-Mvi was not active in fully grown oocytes. However, a clear breakdown of head vesicles in some Oo-Mvi (Fig. 1e arrowheads and Supplementary Movie 3) was detected in living oocytes, showing complex cellular behaviors occurred in those Oo-Mvi.

Therefore, our imaging results showed that there are two types of communicating structures in the ZP: highly dense TZPs derived from GCs and sparse microvilli with vesicles formed by oocytes. Unlike the well-observed GC-TZPs, Oo-Mvi exhibited a series of characteristics and cellular behaviors. Since oocytes play a central role in controlling follicular growth[4], we therefore, hypothesized that Oo-Mvi act as a regulator of ovarian follicle development in female reproduction.

### The microvilli-specific protein RADIXIN was specifically expressed in oocytes.
To clarify our hypothesis and investigate the functional role of Oo-Mvi in folliculogenesis, we isolated the ZP containing Oo-Mvi (Fig. 2a, arrowheads) from oocytes and performed mass spectrometry (MS) analysis to identify potential regulators of the formation of Oo-Mvi. In total, 66 proteins were detected in ZP samples, including the well-studied ZP structure proteins ZP1/2/3[26] and gap junction protein alpha 1 (GJA1)[27], fertilization-related protein CD9[12], and cytoskeletal proteins β-actin and myosin heavy chain 9 (MYH9)[27] (Fig. 2b).

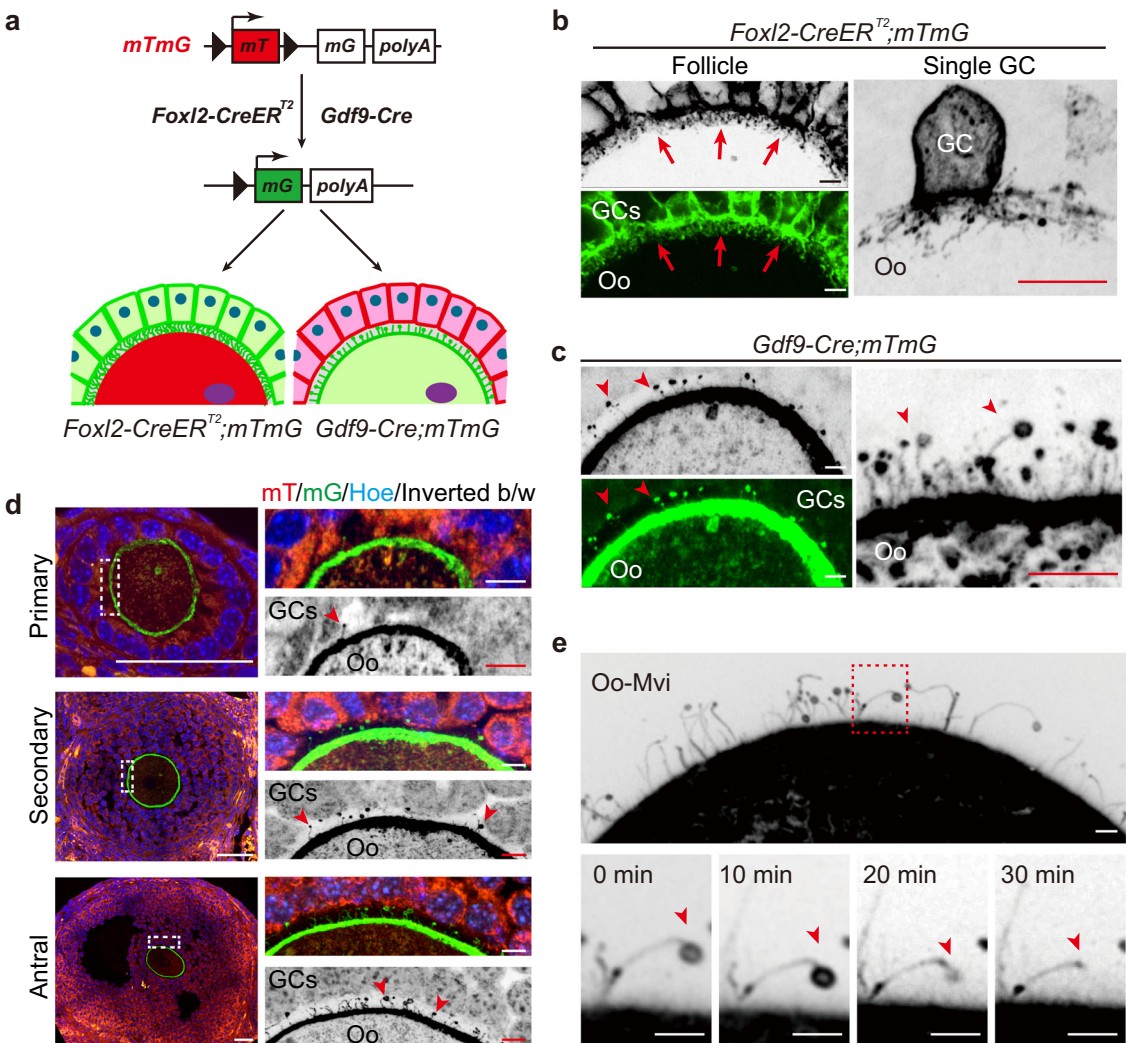

**Fig. 1 Imaging cell-specific communicating structures in the ZP by endogenous-fluorescent labeling in follicles. a** Illustration of the strategy to label the derivation of the cellular communicating structures in the ZP by cellular specifically expressed CreER[T2] or Cre recombinase. Membrane-localized red-fluorescent protein (mT) switches to green-fluorescent protein (mG) in GCs of *Foxl2-CreER[T2];mTmG* follicles and oocytes of *Gdf9-Cre;mTmG* follicles. **b** Images of *Foxl2-CreER[T2];mTmG* ovarian sections showing the high density of GC-TZPs (arrows), and labeling of a single GC showing the messy tree root-like structure of GC-TZPs in detail (right). **c** Images of *Gdf9-Cre;mTmG* ovarian sections, the mushroom-like Oo-Mvi (arrowheads) distributed in a low density (left) and high magnification showing the vesicle tips (arrowheads) of Oo-Mvi (right). **d** The developmental dynamics of Oo-Mvi (arrowheads), showing the establishment of Oo-Mvi accompanied by follicle growth. **e** Time-lapse imaging of Oo-Mvi in living oocytes showing the breakdown of vesicles (arrowheads) of Oo-Mvi (Supplementary Movie 3). All experiments were repeated at least three times, and representative images are shown. Oo oocyte. GCs granulosa cells. ZP zona pellucida. **b–e** The colors were inverted to black/white (b/w) to highlight GC-TZPs (**b**) and Oo-Mvi (**c–e**). Scale bars: 5 μm (**b–d**) and 2 μm (**e**).

Furthermore, one ERM family member, RADIXIN (RDX), which has been reported to control the formation of microvilli in many organs[28], was also detected in our MS results (Fig. 2b and Supplementary Fig. 2). To confirm our findings, we performed quantitative real-time PCR (QRT-PCR) to detect the mRNA levels of different ERM family members in the ovary. The mRNA level of *Rdx* in oocytes was approximately 700 times higher than that in ovarian somatic cells and other tissues (Fig. 2c), whereas the mRNA of the other two ERM family members, *Ezrin* and *Moesin*, were not highly expressed in the oocytes (Supplementary Fig. 3a, b). At the protein level, western blot results also showed that RDX and its activated form phosphorylated (p-) ERM were highly expressed in oocytes (Fig. 2d). These results showed that RDX might be a specific microvillus-related protein in oocytes that contributes to the formation of Oo-Mvi in oocyte development.

We next detected the protein localization of RDX as well as other ERM family members in the ovary. Consistent with the QRT-PCR result, EZRIN and MOESIN were only detected in non-follicular cells such as stromal cells or blood vessels in the ovary (Supplementary Fig. 3c, arrows), and no expression was found in oocytes (Supplementary Fig. 3c, arrowheads). In sharp contrast, the immunofluorescent signals of RDX were detected in both the membrane and cytoplasm of oocytes at all stages of folliculogenesis (Fig. 2e, arrows and arrowheads), including dormant primordial follicles (Fig. 2e, arrowheads). Meanwhile, the localization of RDX in Oo-Mvi under high magnification (Fig. 2f, arrowheads) indicated that RDX might participate in the formation of Oo-Mvi. Compared to total RDX, activated p-ERM was only localized on the cell membrane (Fig. 2g, arrows) and Oo-Mvi in growing oocytes (Fig. 2h, arrowheads) but was not observed in dormant oocytes of primordial follicles (Fig. 2g,

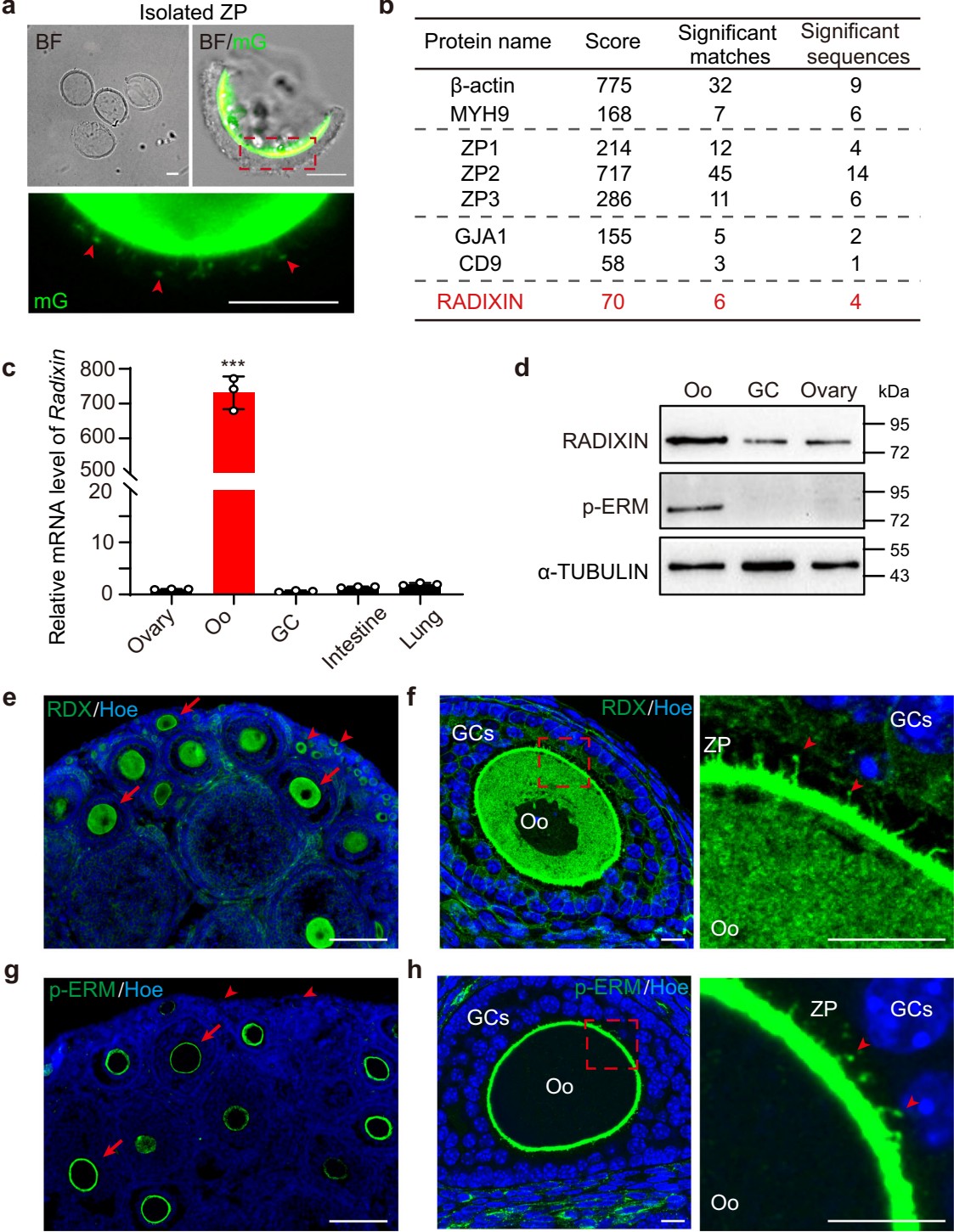

**Fig. 2 The microvilli-related protein RDX was specifically expressed in oocytes. a** Isolated ZP from *Gdf9-Cre;mTmG* oocytes with Oo-Mvi (arrowheads) kept in the samples for MS analysis. **b** MS analysis revealing the expression of RDX in the isolated ZP. **c** Relative mRNA levels of *Rdx* in different follicle components and tissues showing high *Rdx* expression in oocytes. *n* = 3 and *P*-value (Ovary vs. Oo) = 0.0014. **d** Western blot showing that RDX and p-ERM are highly expressed in oocytes. **e** Immunostaining showing RDX protein localized in the oocytes of follicles from the dormant primordial (arrowheads) to growing (arrow) stages. **g** p-ERM highly expresses on the membrane of oocytes in growing follicles (arrows) but not on that of dormant oocytes in primordial follicles (arrowheads). **f, h** High-magnification images showing the localization of RDX (**f**) or p-ERM (**h**) in the Oo-Mvi (arrowheads). All experiments were repeated at least three times, and representative images are shown. Data are presented as the mean ± SD with experiments performed in triplicate. Data were analyzed by two-tailed unpaired Student's *t*-test and ***P < 0.001. Oo oocyte. GCs granulosa cells. ZP zona pellucida. Scale bars: 25 μm (**a**), 100 μm (**e, g**), 10 μm (**f, h**).

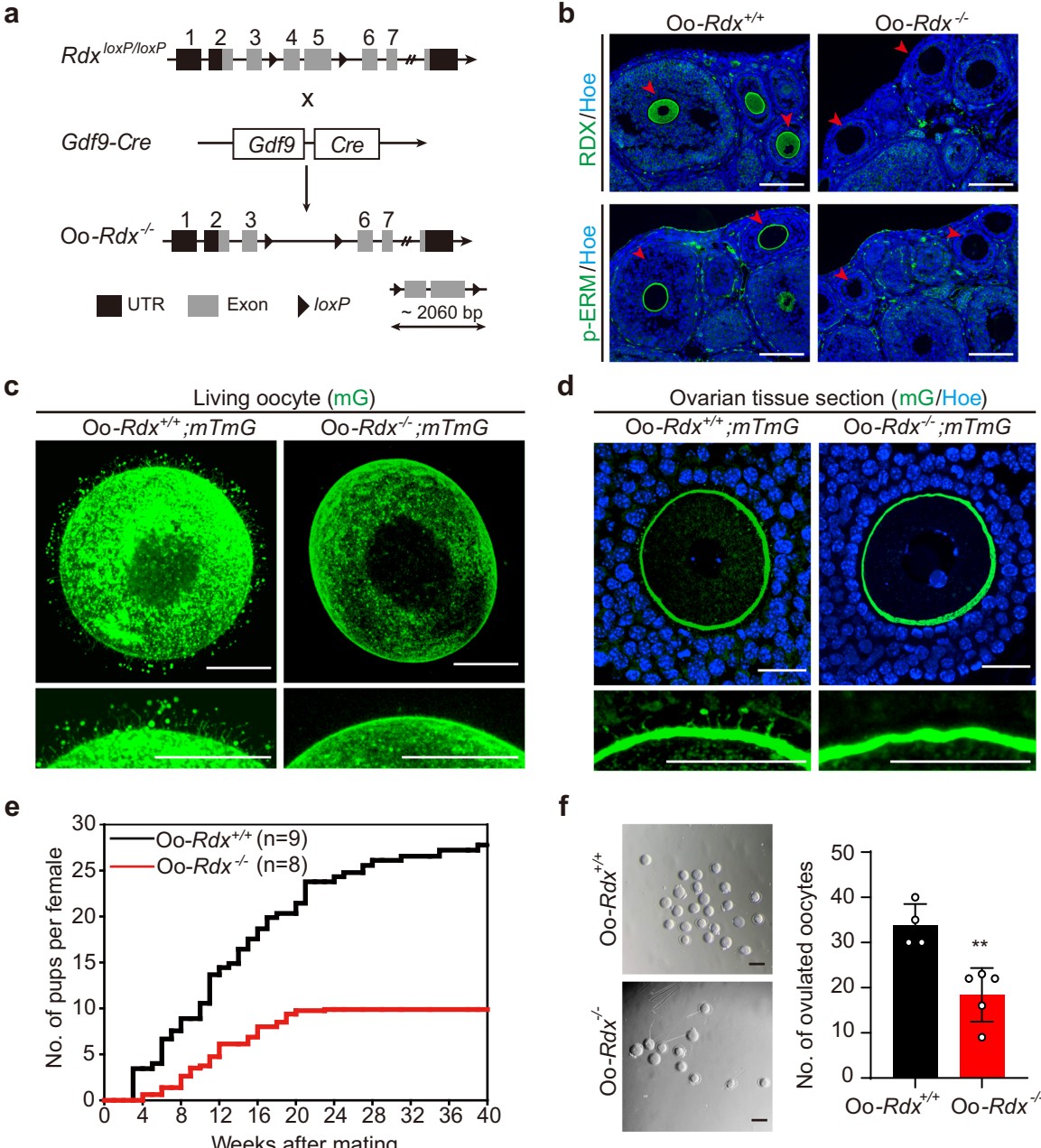

**Fig. 3 Deletion of Rdx in oocytes led to a failure of Oo-Mvi formation and shortened reproductive lifespan in females. a** Schematic representation of the deletion of *Rdx* exons 4 and 5 by *Gdf9-Cre*-mediated recombination in oocytes of Oo-*Rdx*$^{-/-}$. **b** Immunofluorescence detection of RDX and p-ERM showing successful deletion of RDX in oocytes (arrowheads) of Oo-*Rdx*$^{-/-}$. $n = 6$ ovaries per group. **c, d** Imaging the living oocytes (**c**) and ovarian tissue sections (**d**) showing a failure of the construction of Oo-Mvi on the oocyte surface in Oo-*Rdx*$^{-/-}$;mTmG females. $n = 40$ Oo-*Rdx*$^{+/+;mTmG}$ and 16 Oo-*Rdx*$^{-/-;mTmG}$ oocytes (**c**), 12 Oo-*Rdx*$^{+/+;mTmG}$ and 9 Oo-*Rdx*$^{-/-;mTmG}$ ovaries (**d**). **e** Fertility check showing a subfertility phenotype in the Oo-*Rdx*$^{-/-}$ females with significantly decreased litter size and number and a shortened reproductive lifespan ($n = 8$) compared to that in Oo-*Rdx*$^{+/+}$ mice ($n = 9$) during 40 weeks of mating. **f** A significantly lower ovulated oocyte number in Oo-*Rdx*$^{-/-}$ females ($n = 5$) compared to that in Oo-*Rdx*$^{+/+}$ mice ($n = 4$), indicating that the abnormal development of oocytes caused subfertility in Oo-*Rdx*$^{-/-}$ females. *P*-value = 0.0036. Representative images are shown. Data are presented as the mean ± SD. Data were analyzed by two-tailed unpaired Student's *t*-test and **$P < 0.01$. Scale bars: 100 μm (**b**, **f**), 25 μm (**c**, **d**).

arrowheads). This localization is consistent with the formation of Oo-Mvi in follicle development, indicating that RDX in oocytes is involved in the formation of Oo-Mvi, and the major function of RDX and Oo-Mvi might be related to the growth of oocytes and follicles in the ovary.

**Oocytic *Rdx* deletion resulted in a failure of the formation of Oo-Mvi and a shortened reproductive lifespan in females.** To

study the function of Oo-Mvi, we generated a conditional *Rdx* knockout mouse by flanking exons 4/5 with *loxP* sequences (*Rdx*$^{loxP}$) (Fig. 3a). *Rdx*$^{loxP/loxP}$ females were crossed with *Gdf9-Cre* males to delete *Rdx* in oocytes at the primordial follicle stage (Fig. 3a). The resulting *Gdf9-Cre;Rdx*$^{loxP/loxP}$ mice were referred to as Oo-*Rdx*$^{-/-}$ mice, and the littermates of *No-Cre;Rdx*$^{loxP/loxP}$ females, referred to as Oo-*Rdx*$^{+/+}$, were used as the control. To validate the efficiency of deletion in Oo-*Rdx*$^{-/-}$ mice, we performed immunofluorescent staining analysis to detect the

expression of RDX in the ovaries. In the Oo-$Rdx^{+/+}$ ovary, the expression of both RDX and p-ERM was observed in oocytes of follicles (Fig. 3b, arrowheads). In sharp contrast, neither RDX nor p-ERM expression was detected in oocytes of Oo-$Rdx^{-/-}$ ovaries (Fig. 3b, arrowheads), demonstrating a successful oocytic deletion of *Rdx*.

Since RDX is highly expressed in the cortex of oocytes, we first detected the construction of oocyte cortex in Oo-$Rdx^{-/-}$ females followed a standard approach by labeling the actin with Alex488-phalloidin and detecting the cortical granules with LCA-FITC in oocytes[10,29,30]. As a result, an identical cortex was found in the mutant Oo-$Rdx^{-/-}$ oocytes and Oo-$Rdx^{+/+}$ controls (Supplementary Fig. 4), showing that deletion of *Rdx* had no effect on the cortical formation of oocytes. We next detected the cellular architecture changes of oocytes in Oo-$Rdx^{-/-}$ mice by introducing the *mTmG* reporter mice into the model to outline the cell surface of the gene-modified oocytes (Supplementary Fig. 5). As shown in Fig. 3c, normal Oo-Mvi with vesicles were found on the surface of living oocytes in control Oo-$Rdx^{+/+}$;*mTmG* females. However, a smooth surface with few microvilli was observed on oocytes of Oo-$Rdx^{-/-}$;*mTmG* ovaries, showing that the deletion of RDX successfully disrupted the formation of microvilli on oocytes (Fig. 3c). Meanwhile, histological analysis showed that Oo-Mvi were hardly detected in the ZP of Oo-$Rdx^{-/-}$;*mTmG* follicles, confirming that *Rdx* deletion led to a failure of the establishment of Oo-Mvi during follicle development (Fig. 3d).

We next focused our study on the effect of the loss of Oo-Mvi on female fertility. Surprisingly, the absence of Oo-Mvi did not lead to complete infertility in Oo-$Rdx^{-/-}$ females. Most Oo-$Rdx^{-/-}$ females gave birth from 4 weeks after mating, but all of them became infertile before 24 weeks after mating, showing a phenotype of a remarkably shortened reproductive lifespan compared to their Oo-$Rdx^{+/+}$ littermates (Fig. 3e). Moreover, the time of first labor was delayed, and the litter frequency and litter size were significantly decreased in Oo-$Rdx^{-/-}$ females compared to those in the control mice (Fig. 3e), demonstrating a subfertile phenotype in female Oo-$Rdx^{-/-}$ mice. Meanwhile, Oo-$Rdx^{-/-}$ females only ovulated $18.40 \pm 2.66$ oocytes after superovulation, which was in sharp contrast to the $33.75 \pm 2.39$ ovulated oocytes in Oo-$Rdx^{+/+}$ females (Fig. 3f), indicating that abnormal folliculogenesis is the major cause of female subfertility in Oo-$Rdx^{-/-}$ females.

**Loss of Oo-Mvi led to a retardation of oocyte and follicle development in Oo-$Rdx^{-/-}$ females.** To investigate the inner mechanisms of female subfertility in Oo-$Rdx^{-/-}$ mice, we first detected the development of Oo-$Rdx^{-/-}$ ovaries at different ages (Fig. 4a and Supplementary Fig. 6). At Postnatal Day (PD) 5, a similar morphology of ovaries was found in Oo-$Rdx^{-/-}$ mice and control mice (Supplementary Fig. 6a), showing that oocyte deletion of *Rdx* does not affect the formation or survival of dormant primordial follicles. At PD35, although the size of ovaries did not differ between Oo-$Rdx^{-/-}$ and control females, a remarkably lower density of follicles was observed in Oo-$Rdx^{-/-}$ ovaries than in the Oo-$Rdx^{+/+}$ female littermates (Fig. 4a, PD35). At 2 months, the ovaries in mutant females were significantly smaller than those in the controls (Figs. 4a, 2 months), and more atresia follicles were observed in the Oo-$Rdx^{-/-}$ ovaries (Fig. 4a, arrows and Supplementary Fig. 6b, arrows). Consistent with the phenotype of premature ovarian insufficiency, the Oo-$Rdx^{-/-}$ ovaries no longer displayed normal ovarian morphology at 8 months (Fig. 4a, 8 months), and no corpus luteum but many atresia follicles (Fig. 4a, arrows) were observed in the mutant ovaries. Quantification of follicles in Oo-$Rdx^{-/-}$ and control females confirmed the accelerating follicle loss in Oo-$Rdx^{-/-}$ ovaries (Fig. 4b), and the peak of follicle loss occurred in Oo-$Rdx^{-/-}$ ovaries before adulthood (Fig. 4b), which is a period of fast activation

and growth of ovarian follicles in the female reproductive lifespan[25]. Therefore, we hypothesized that loss of Oo-Mvi might decrease the developmental capability of growing follicles in ovaries, leading to increased follicle loss and premature ovarian insufficiency in females.

To test our hypothesis, we isolated and cultured ovarian cortical pieces or follicles from Oo-$Rdx^{-/-}$ and Oo-$Rdx^{+/+}$ females and traced the developmental dynamics of growing oocytes and follicles in vitro. By culturing the ovarian cortical pieces from Oo-$Rdx^{-/-}$;*mTmG* and Oo-$Rdx^{+/+}$;*mTmG* females, we found significant retardation of the oocyte diameter increase in Oo-$Rdx^{-/-}$;*mTmG* females compared to that in controls during 7 days of in vitro growth (Fig. 4c, d, $1.28 \pm 0.09$ times vs. $1.79 \pm 0.07$ times), showing that loss of Oo-Mvi by *Rdx* deletion disrupted the growth of oocytes in the follicles. Next, we isolated and cultured follicles from Oo-$Rdx^{-/-}$ ovaries and Oo-$Rdx^{+/+}$ ovaries. Consistent with the retardation of oocyte growth, the increase in follicle diameters in Oo-$Rdx^{-/-}$ females ($117.60 \pm 1.50$ to $248.50 \pm 15.25$ μm) was also significantly slower than that in Oo-$Rdx^{+/+}$ females ($121.90 \pm 1.39$ to $322.50 \pm 21.79$ μm) during 8 days of in vitro culture (Fig. 4e, f). These results clearly showed that loss of Oo-Mvi might decrease the developmental capability of growing follicles in ovaries.

**Loss of Oo-Mvi affected the development of follicle GCs in Oo-$Rdx^{-/-}$ females.** Based on the behaviors of Oo-Mvi in living oocytes and the general phenotypes of Oo-$Rdx^{-/-}$ females, we proposed that the function of Oo-Mvi might be to regulate the development of GCs that support the growth of follicles. To test our hypothesis, we detected the developmental status of GCs in Oo-$Rdx^{-/-}$ females. Although GC proliferation rate varies greatly among follicles, we found that the rate of GC proliferation in primary and secondary follicles was significantly lower in Oo-$Rdx^{-/-}$ females than in controls (Fig. 5a, arrows and b, $14.92\% \pm 2.53\%$ vs. $34.74\% \pm 2.52\%$ in the primary stage, $34.47\% \pm 2.17\%$ vs. $43.83\% \pm 2.65\%$ in the secondary stage). Meanwhile, TUNEL detection also showed a remarkable increase in GC apoptosis in primary and secondary follicles of Oo-$Rdx^{-/-}$ ovaries (Fig. 5c, arrows and d, $16.44\% \pm 3.52\%$ vs. 0 in the primary stage, $29.98\% \pm 4.87\%$ vs. $13.51\% \pm 0.89\%$ in secondary stage). These results showed that loss of Oo-Mvi decreased the developmental capability of GCs in early growing follicles in ovaries.

To further investigate the effect of the loss of Oo-Mvi on the development of GCs, we next tested the formation of GC-TZPs in oocytes lacking Oo-Mvi in Oo-$Rdx^{-/-}$ females. Oocytes were collected from Oo-$Rdx^{-/-}$ or Oo-$Rdx^{+/+}$ ovaries at PD35 and F-actin was stained to show the TZPs within the ZP[10]. As expected, a high density of TZPs covering the oocyte surface was observed in the ZP of control oocytes (Fig. 5e, Oo-$Rdx^{+/+}$). However, only a small portion of GCs containing TZPs existed as groups in the ZP (Fig. 5e, Oo-$Rdx^{-/-}$, arrows), and significantly fewer GC-TZPs on the surface of oocytes were detected in Oo-$Rdx^{-/-}$ females than in controls (Figs. 5f, $28.14 \pm 3.94$ vs. $121.70 \pm 9.03$). This result showed a failure of GC-TZP network construction in the follicles without Oo-Mvi, indicating a controlling role of Oo-Mvi in the formation of GC-TZPs.

Previous studies reported that OSFs, such as GDF9, control the formation of GC-TZPs[10]. To further investigate whether the effect of the loss of Oo-Mvi on follicle development is related to OSF release, we cultured ovarian cortical pieces from Oo-$Rdx^{-/-}$ females with or without GDF9 and examined the formation of GC-TZPs. After 24 h of culture with GDF9 (100 ng/mL), the failure of TZP network construction was well rescued by GDF9 supplementation (Fig. 5g, Oo-$Rdx^{-/-}$+GDF9). TZP-counting results confirmed our observations that supplying GDF9 significantly increased the density of GC-TZPs in Oo-

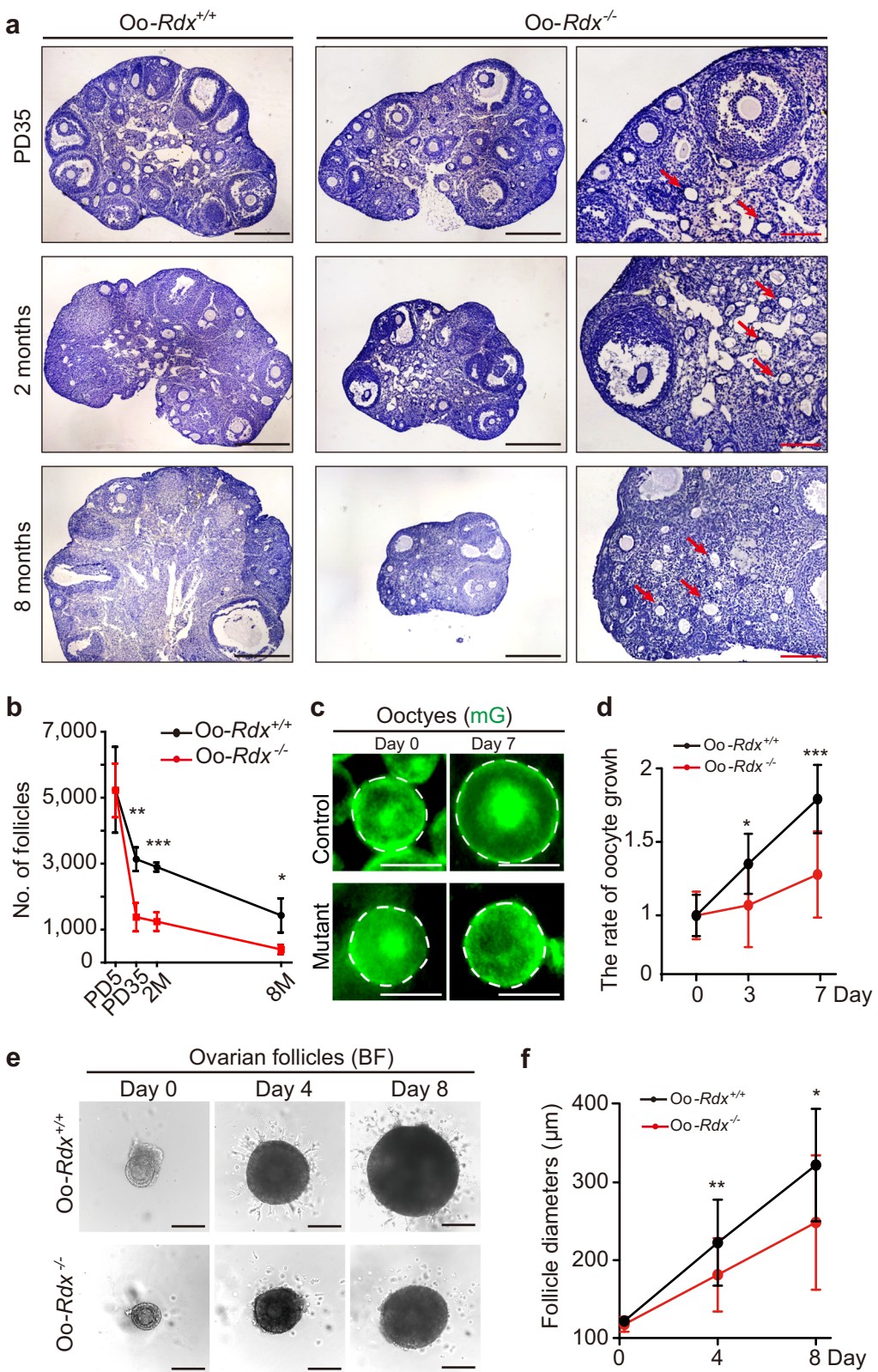

$Rdx^{-/-}$ follicles (Fig. 5h, $60.05 \pm 5.72$ in Oo-$Rdx^{-/-}$+GDF9, compared to $26.32 \pm 3.00$ in Oo-$Rdx^{-/-}$, and $69.25 \pm 4.42$ in Oo-$Rdx^{+/+}$). Moreover, we found a significantly decreased expression of *Ptgs2* and *Tnfnip6*, which were reported as the GDF9-downstream genes[18] in Oo-$Rdx^{-/-}$ females (Supplementary Fig. 7), indicating deletion of *Rdx* led to a GDF9 related deficiency of ovaries. These results supported our hypothesis that

Oo-Mvi regulate GC function and follicle development, and this regulation relates to the release of OSFs in the ovary.

**Oo-Mvi were involved in the release of OSFs to manipulate the development of follicles.** Our in vivo and in vitro experimental evidence strongly indicated that Oo-Mvi might regulate OSF

**Fig. 4 The failure of Oo-Mvi formation in Oo-$Rdx^{-/-}$ females led to abnormal folliculogenesis. a** Histological analysis of the ovarian morphological changes in Oo-$Rdx^{-/-}$ females, showing a decrease in ovarian size and an increase in follicle death (arrows) in the mutant ovaries. **b** Follicle counting confirmed a fast follicle loss in Oo-$Rdx^{-/-}$ ovaries. P-value: 0.98 (PD5, $n = 3$), 0.0015 (PD35, $n = 5$), 0.0008 (2 M, $n = 3$), 0.03 (8 M, $n = 3$). **c, d** Tracing the growth of oocytes in cultured ovarian pieces of Oo-$Rdx^{+/+}$;mTmG and Oo-$Rdx^{-/-}$;mTmG (**c**). A significant retardation of oocyte growth in Oo-$Rdx^{-/-}$;mTmG ($n = 11$) ovaries compared to that in Oo-$Rdx^{+/+}$;mTmG ($n = 11$) ovaries (**d**). P-value: 0.015 (day 3), 0.0002 (day 7). **e, f** Tracing the growth of ovarian follicles from Oo-$Rdx^{-/-}$ and Oo-$Rdx^{+/+}$ ovaries in vitro (**e**), showing significant retardation of the increase in follicle diameter in Oo-$Rdx^{-/-}$ mice ($n = 9$, red) compared to that in controls ($n = 6$, black). P-value: 0.0057 (day 4), 0.014 (day 8) (**f**). Representative images are shown. Data are presented as the mean ± SD. Data were analyzed by two-tailed unpaired Student's $t$-test and ***$P < 0.001$, **$P < 0.01$, *$P < 0.05$. Scale bars: 500 μm (black) and 100 μm (red) (**a**), 25 μm (**c**), and 200 μm (**e**).

release to control the crosstalk between oocytes and GCs. To confirm our hypothesis, the expression of GDF9 was detected as a representative molecule in the oocytes of PD35 ovaries. By co-staining of RDX and GDF9 in ovaries, we found that the signal of GDF9 was clearly observed in the vesicles of Oo-Mvi on oocytes (Fig. 6a, arrowheads, Supplementary Fig. 8). This result showed that GDF9 distributed in the tip of Oo-Mvi, indicating OSFs were transported to the vesicles of Oo-Mvi to be released.

To clarify the direct relationship between Oo-Mvi and OSF release, we next performed oocyte microinjection to introduce GDF9 coupled with red-fluorescent rhodamine (R-GDF9) using the NHS-rhodamine amine-specific labeling method. In this system, the localization of injected R-GDF9 in oocytes could be directly visualized in culture. As the negative control, rhodamine-labeled bovine serum albumin (R-BSA) (50 μg/mL) was first injected into oocytes from wild-type females, and a dispersed red-fluorescent signal (Fig. 6b, R-BSA) showed a uniform distribution of non-specific protein in the oocytes. In sharp contrast, the R-GDF9 (50 μg/mL) signal aggregated as cloudy structures in the cytoplasm of oocytes (Fig. 6b, R-GDF9), and various R-GDF9 spots were observed in the ZP of injected oocytes (Fig. 6b, R-GDF9, arrowheads, Supplementary Movies 4 and 5), which were similar to the distribution of the vesicles of Oo-Mvi in Gdf9-Cre;mTmG oocytes (Supplementary Fig. 9, arrows). Interestingly, we found a clear reduction of Rho-fluorescent intensity in the oocytes with the increase of time after R-GDF9 injection from 15 to 30 min (Fig. 6b, c), showing a fast release of the R-GDF9 from oocytes. With the decrease of the R-GDF9 signal in oocytes, a time-dependent increase of the R-GDF9 spot number (Fig. 6b, arrowheads, Supplementary Movies 4 and 5) was observed on oocytes after R-GDF9 injection. As we showed in Fig. 6d, the average number of R-GDF9 spots was increased 8 ± 4 at ~15 min to 25 ± 9 at ~30 min after injection. Therefore, these results supplied direct experimental evidence showing that the injected exogenous OSF was enriched and released through the Oo-Mvi.

Since the secretory proteins including the OSFs are transported through the endoplasmic reticulum (ER) to the plasma membrane[31], we next detected the localization of ER in GV oocytes by the ER-tracker with a high-resolution approach. We found that the ER exhibited an aggregative and cloudy distribution in the oocyte cytoplasm (Fig. 6e, top panel), which was consistent with the previously reports[32,33]. Interestingly, we also found lots of ER-tracker labeled bubbles (ER bubbles) in the ZP (Fig. 6e, top panel, arrowheads), which were similar as the distribution of Oo-Mvi and the R-GDF9 bubbles (Fig. 6b, e, arrowheads and Supplementary Movie 6). The counting results showed that a comparable number of ER bubbles (107 ± 6) and mG-labeled vesicles (110 ± 7) on Oo-Mvi of Gdf9-Cre;mTmG ovaries (Fig. 6f). Furthermore, the quantitation of the fluorescent intensity showed that these ER bubbles exhibited a significantly higher signal intensity than that in the oocyte cytoplasm (Fig. 6g and Supplementary Fig. 10). Therefore, these results supplied evidence suggesting the enrichment of ER-transported protein in

the vesicles of Oo-Mvi, which was in consistent with our conclusion about the OSF release function of Oo-Mvi.

Finally, we detected the mRNA levels of several well-studied OSFs, including *Gdf9*, *Bmp15*, and *Fgf8*[34], in fully grown oocytes from Oo-$Rdx^{-/-}$ and Oo-$Rdx^{+/+}$ ovaries. We found that the mRNA levels of these OSFs in Oo-$Rdx^{-/-}$ oocytes were significantly higher (*Gdf9*, 2.33 ± 0.28 times, *Bmp15*, 2.20 ± 0.38 times, and *Fgf8*, 2.91 ± 1.54 times) than those in the Oo-$Rdx^{+/+}$ control (Fig. 6h). Since the majority of oocyte mRNA transcripts occur at the early stage of follicle development[35], the increased levels of mRNAs in ovulated oocytes showed that the consumption of storage OSF mRNAs was dramatically reduced in oocytes without Oo-Mvi. Therefore, the data confirmed our finding that Oo-Mvi contributed to the release of OSFs.

Taken together, our data provide direct evidence that oocytes establish a microvillus network to control the development of GCs in follicles. These Oo-Mvi shorten the physical distance between oocytes and surrounding GCs, and enrich and release OSFs in their vesicle tips to maintain normal follicle development and survival in the ovary. The loss of Oo-Mvi in Oo-$Rdx^{-/-}$ females decrease the efficiency of OSF stimulation, thus leading to abnormal GC development and finally affecting the survival of follicles and shortening the reproductive lifespan in females (Fig. 6i).

## Discussion

The developing regulation of oocytes is a complex and multi-cell participating process in all vertebrate species. In mammals, each oocyte is covered by surrounding somatic GCs to form a relatively independent structure called ovarian follicles as the fundamental unit of female reproduction[3]. Active bidirectional communications between oocytes and GCs are essential for oogenesis in adult ovaries, and oocytes have been well accepted to dominate intercellular conversation during folliculogenesis[7]. By timely and orderly secretion of the oocyte-secreted factors, oocytes govern the proliferation and differentiation of GCs to intermediate the germ-somatic crosstalk[4,36]. Controversially, oocyte also formed a glycoprotein egg coat, called the zona pellucida (ZP) to block the direct communication with the follicle somatic cells[5]. Till now, the accurate cellular constructs on oocytes supporting these fine-regulated dialogs are unclear. Here, by combining endogenous-fluorescent mouse models with a high-resolution imaging system, we identified and imaged two types of microvillus structures in detail in the mouse ZP, i.e., GC-TZPs and Oo-Mvi. Moreover, we supply direct evidence showing that oocytes forming Oo-Mvi act as a cellular antenna to regulate OSF release, therefore maintaining the proper reproductive lifespan in females.

At the single-cell level, our data showed that each GC in the inner layer of follicles establishes a high-density GC-TZP network to connect to oocytes directly. This observation is consistent with previous views that GC-TZPs support the growth of oocytes[2,36]. Meanwhile, our results confirm previous findings that the formation of GC-TZPs is controlled by the secretion of OSFs (such

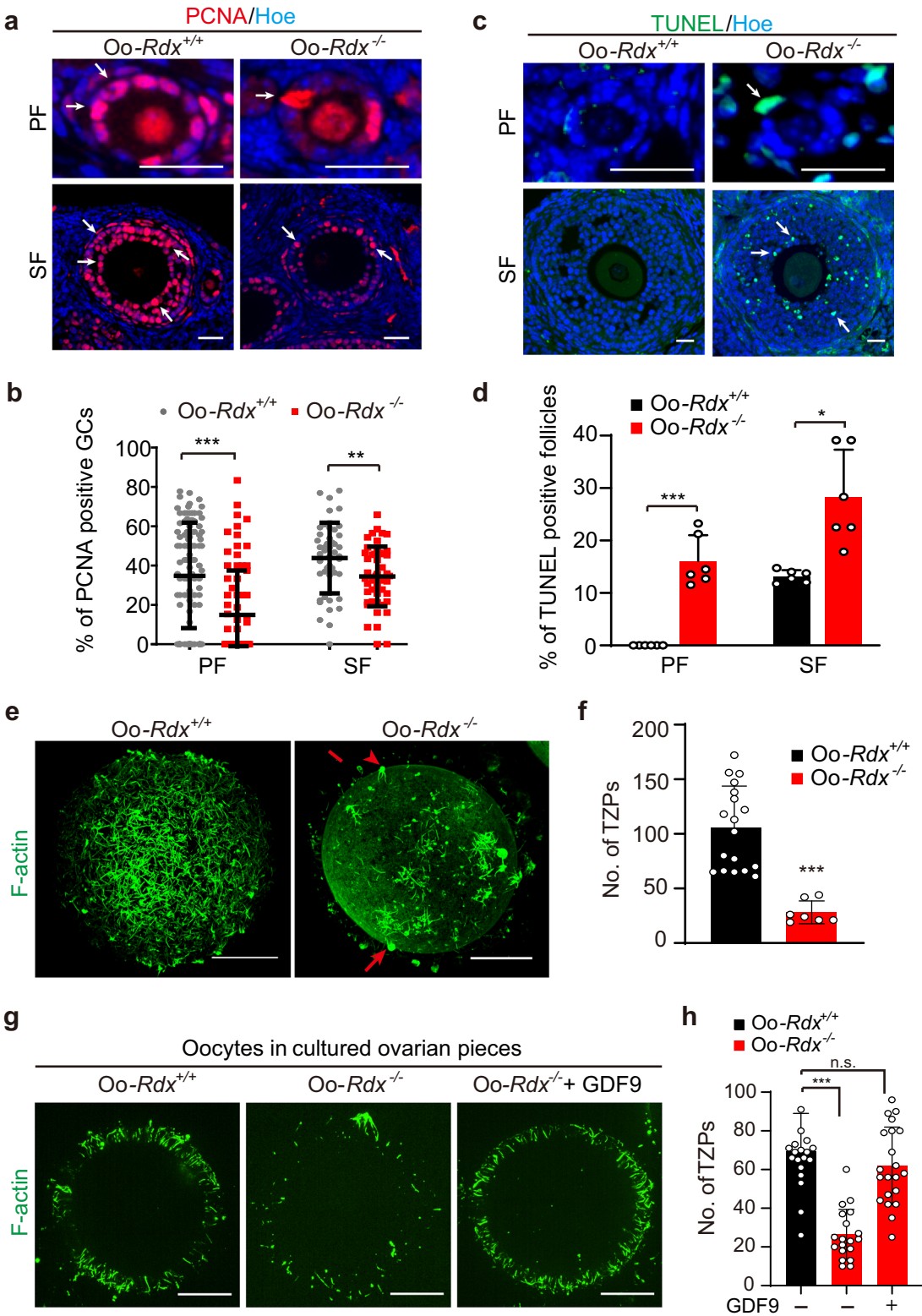

as GDF9) from oocytes[10,36]. Therefore, these findings further support the classical theory that oocytes govern the growth of follicles in mammals.

Consistent with previous studies that demonstrated that oocytes also form the communicating structures in different species such as surf clams[37], frogs[38], rodent[39–41], bovine[42–44], and humans[45,46], our results clearly described the oocyte-derived microvilli during the process of mammalian oogenesis. Compared to GC-TZPs, our

high-resolution living cell observation of Oo-Mvi with endogenous-fluorescent labeling showed the oocyte-derived communicating structures exhibit several interesting and previously undefined cellular characteristics, supplying important indications for the functional role of such Oo-Mvi. First, the Oo-Mvi, at a relatively low density, is not directly connected to the GC membrane, indicating that these structures do not act as channels for nutrient transport. Moreover, the fact that the majority of Oo-Mvi

**Fig. 5 Loss of Oo-Mvi disrupted OSF-regulated GC development. a, b** PCNA staining (arrows) of GC proliferation (**a**), showing a significant decrease in GC proliferation in the early growing follicles of Oo-$Rdx^{-/-}$ ovaries (**b**). Each point represents the GC-PCNA rate in a single follicle (PF: control $n = 105$, Oo-$Rdx^{-/-}$ $n = 77$, P-value = 0.0000011; SF: control $n = 46$, Oo-$Rdx^{-/-}$ $n = 49$, P-value = 0.0072). **c, d** TUNEL detection (arrows) of apoptosis (**c**), showing a significantly increased ratio of follicles with TUNEL-positive GCs in Oo-$Rdx^{-/-}$ ovaries ($n = 6$). P-value = 0.0007 (PF), 0.02 (SF) (**d**). **e** F-actin staining GC-TZPs (arrows) showing a failure of GC-TZP construction in Oo-$Rdx^{-/-}$ oocytes. **f** Quantification of GC-TZPs confirmed a dramatically decreased density of GC-TZPs in Oo-$Rdx^{-/-}$ oocytes ($n = 18$) compared to that in Oo-$Rdx^{+/+}$ controls ($n = 7$). P-value = 0.0000008. **g** Supplying GDF9 rescued the defect of GC-TZP formation in Oo-$Rdx^{-/-}$ ovaries. **h** Quantification of GC-TZPs in Oo-$Rdx^{+/+}$ ($n = 20$), Oo-$Rdx^{-/-}$ ($n = 19$), and Oo-$Rdx^{-/-}$+GDF9 ($n = 21$) oocytes, showing the successful rescue of GC-TZP defect by GDF9 supplementation. P-value: 0.000000027 (Oo-$Rdx^{+/+}$ vs. Oo-$Rdx^{-/-}$); 0.21 (Oo-$Rdx^{+/+}$ vs. Oo-$Rdx^{-/-}$+GDF9). Data are presented as the mean ± SD. PF primary follicle, SF secondary follicle. Representative images are shown. Data were analyzed by two-tailed unpaired Student's *t*-test and ***$P < 0.001$, **$P < 0.01$, *$P < 0.05$, n.s. $P \geq 0.05$. Scale bars: 25 μm.

contained vesicle heads and their breakdown indicates an exocrine function of Oo-Mvi in the regulation of folliculogenesis. Indeed, our functional study strongly supports the idea that Oo-Mvi act as a regulating structure since the failure of Oo-Mvi formation leads to the retardation of oocyte and follicle growth, and accelerated follicle loss in the ovary. Interestingly, although our results showed that Oo-Mvi play an important role in folliculogenesis, we found that the formation of Oo-Mvi is not indispensable for the survival of oocytes and follicles in vivo. Even though the construction of Oo-Mvi is destroyed by *Rdx* deletion, some of the oocytes can still develop to ovulation in young adulthood. Therefore, the mutant females are fertile but have a shortened reproductive lifespan, just like the clinical symptom in patients with premature ovarian insufficiency[47]. With our experimental data, we proposed that the role of Oo-Mvi is an optimized system to select the qualified follicles, especially the early growing follicles. Meanwhile, although our results showed that OSFs are released through Oo-Mvi, there is no clear evidence to show a lack of secreted GDF9 in Oo-$Rdx^{-/-}$ females. Therefore, we also hypothesized that the release of OSFs does not completely rely on Oo-Mvi since the folliculogenesis never passed the secondary stage in *Gdf9*$^{-/-}$ females[17]. Also, inconsistency of the phenotypes between *Gdf9*$^{-/-}$ and Oo-$Rdx^{-/-}$ females implied complex underlying mechanisms of either Oo-Mvi forming regulations or the OSF release controlling on oocytes for future study. Interestingly, a previous study reported that oocytes also form membrane fusion with GCs to mediate the secretion of GDF9[48], and these observations further support our hypothesis that Oo-Mvi play a regulating role in controlling folliculogenesis. Meanwhile, similar structures of Oo-Mvi were reported on the membrane of cleaved bovine eggs at 2 or 3 cells stage[49], implying that this kind of Mv might be a conserved structure during oocyte and early embryo development. Therefore, we believed that the major function of Oo-Mvi are to make the release of OSFs timely and orderly, guaranteeing a precise and efficient regulation to determine the developmental fate of ovarian follicles.

Taken together, the observations in the current study identify an elegant microvillus system in the ZP of growing follicles that maintains proper folliculogenesis in mice. After the awakening of dormant follicles, oocytes produce ZP proteins to establish the egg coat, which is essential for oocyte development and maturation[26]. At the same time, RDX-related Oo-Mvi form and elongate on the surface of oocytes to shorten the physical distance between oocytes and GCs. With the formation of Oo-Mvi, OSFs are transported to the tips of Oo-Mvi to construct OSF-rich vesicles. Although the mechanism by which OSFs are transported to the vesicles in Oo-Mvi is still unknown, as these vesicles breakdown, the soluble OSFs are efficiently released to the adjacent space of the ZP. In response to OSF stimulation, a series of cellular events take place in GCs, including the increase of proliferation, the decrease of apoptosis, and the construction of GC-TZPs, which connect with the oocyte membrane to finally support the robust development of oocytes.

## Methods

**Mice.** C57BL/6 mice were from the Laboratory Animal Center of the Institute of Genetics (Beijing, China). *Gdf9-Cre*, *Foxl2-CreER*$^{T2}$, and *mTmG* mice were generated as previously reported[23–25]. *Foxl2-CreER*$^{T2}$ mice were a gift from Dr. Liu Kui. The *Rdx*$^{loxP/loxP}$ and *Gdf9*$^{-/-}$ mice were generated using CRISPR/Cas9-mediated genome engineering by the Model Animal Research Center of Nanjing University (MARC), Nanjing BioMedical Research Institute of Nanjing University, Nanjing, China. Generally, exons 4 and 5 of the *Rdx* gene were selected as the targeted region to delete in the *Rdx*$^{loxP/loxP}$ mice, and the exon 2 of *Gdf9* gene was deleted in the *Gdf9*$^{-/-}$ mice. All mutant mouse strains were on the C57BL/6 background.

In the *Foxl2-CreER*$^{T2}$;*mTmG* females, upon a single injection of tamoxifen (75648, Sigma-Aldrich) at a dosage of 5 μg/kg body weight (BW) at PD8, CreER$^{T2}$ recombinase-mediated recombination to label a single GC after 1 week[50]. With tamoxifen at a high dosage of 20 μg/kg BW for three injections at PD8/10/12, almost all GCs were labeled after 1 week of treatment[25,50].

All mice were housed in mouse facilities under 16/8-h light/dark cycles at 26 °C and humidity 40–70% with access to chow and water at libitum. The animal experiments conformed to the guidelines and regulatory standards of the Institutional Animal Care and Use Committee of China Agricultural University, No. AW8012020-2-3.

**Histological section staining and follicle counting.** For morphological analysis and fluorescent detection at the tissue level, ovarian samples were fixed in 4% paraformaldehyde (PFA, Santa Cruz, 30525-89-4) for 8 h, embedded in paraffin, and sectioned serially at 5 μm. To count the follicle number, tissue sections were stained with hematoxylin (Santa Cruz, sc-24973A). Primordial follicles were counted in every fifth section and multiplied by five to calculate the number of all primordial follicles in each ovary. The growing follicles were counted by scanning all sections and only the follicles with clear oocyte nuclei were counted to exclude the effect of residual structures after oocyte ablation in the ovaries. The number of primordial follicles and growing follicles was summed to the total number of follicles. For fluorescent detection, after deparaffinization, sections were stained with different antibodies and the Hoechst 33342 (Sigma-Aldrich, 14533) was used as a counter-stain to check the cell nucleus. The sections were sealed with an anti-fade fluorescence mounting medium (Applygen, C1210) by coverslips.

**High-resolution imaging of the isolated oocytes to detect the subcellular structure or protein/factor localization.** All high-resolution images of isolated oocytes were acquired by an Andor Dragonfly spinning-disc confocal microscope equipped with a ×63 or ×100, 1.44 N.A. oil objective, a scientific complementary metal-oxide semiconductor (sCMOS) camera (Andor Zyla 4.2), and the 405-nm (Hoe), 488-nm (mG, Alexa Fluor 488-phalloidin, and LCA-FITC) and 568-nm (mT, ER-tracker, and rhodamine) lines of the Andor Integrated Laser Engine (ILE) system with a spinning-disc confocal scan head (Andor Dragonfly 500). Images were acquired by Fusion 2.1 software (https://andor.oxinst.com/products/dragonfly#fusion).

To detect the Oo-Mvi in oocytes, denuded oocytes at GV stages (average diameter around 65 μm) of late secondary or pre-antral follicles were collected by tearing the ovaries with syringe needle from wild type females or Oo-$Rdx^{-/-}$; *mTmG* females and control. The oocytes were moved with mouth pipette to ~20 μL minimum MEMα-FBS-ITS medium: essential medium α (MEMα; Gibco, 32-571-036) with 10% FBS (Gibco, 10-099-141) and 1% insulin-transferrin-selenium (ITS; Sigma-Aldrich, 13146), covered with mineral oil (Sigma-Aldrich, M8410) and photographed in a living cell workstation (Okolab) at 37 °C, 5% CO$_2$. Images were typically acquired at a proper temporal interval (10 min) with an optical slice thickness of 0.5 μm and covered ~40 μm of oocytes.

To observe the cortex actin of oocytes or the GC-TZPs, denuded oocytes at the GV stage (average diameter around 65 μm) were collected from fresh ovaries or GDF9-cultured ovarian pieces of Oo-$Rdx^{+/+}$ and Oo-$Rdx^{-/-}$ females. Then the oocytes were fixed with 4% PFA in PBS for 15 min. After washing with PBS, the fixed oocytes were incubated with Alexa Fluor 488-phalloidin (1:100 dissolved in MEMα-FBS-ITS medium, ThermoFisher Scientific, A12379) with (to detect oocyte

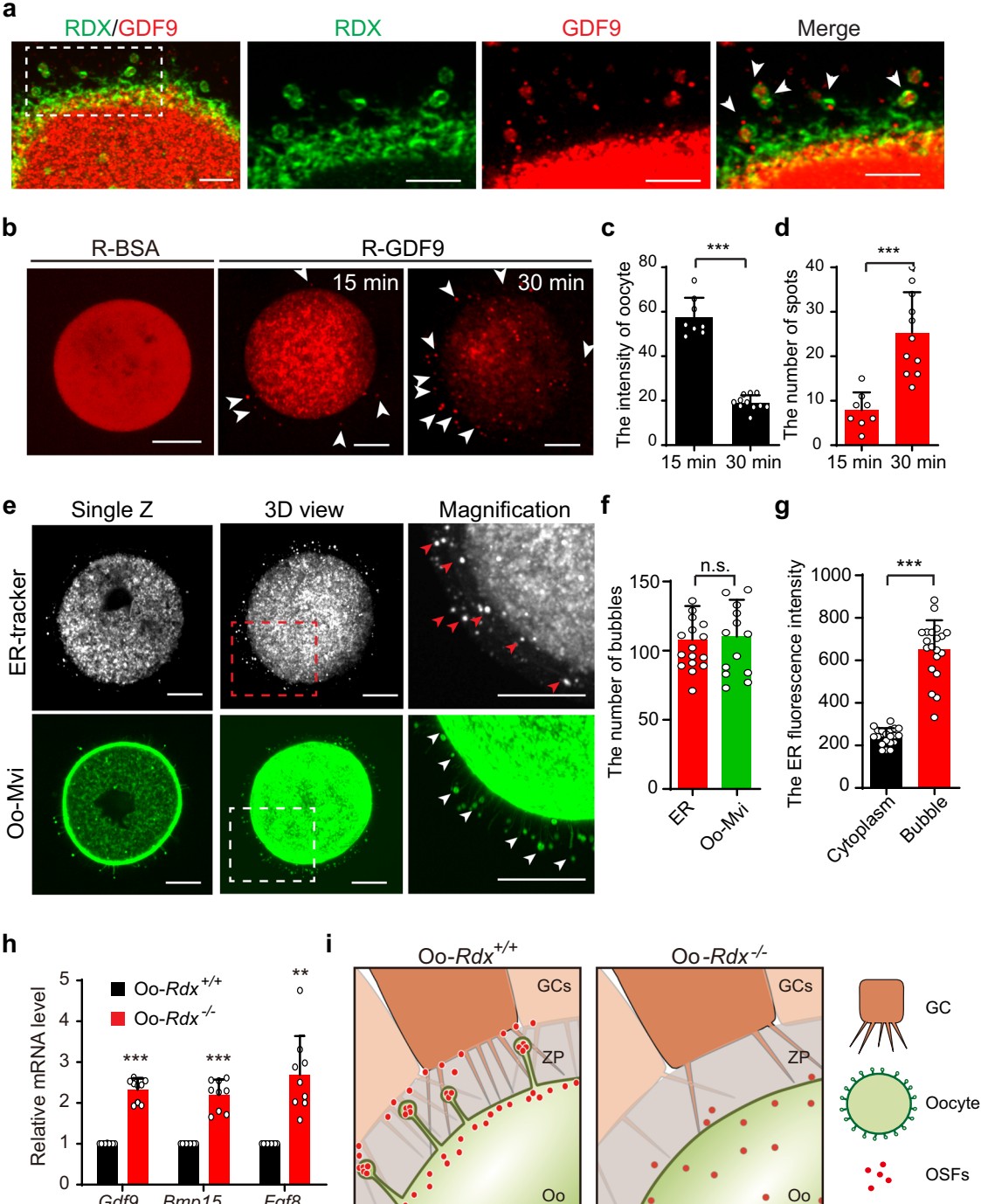

**Fig. 6 OSFs were enriched in Oo-Mvi to manipulate the development of follicles. a** GDF9 was localized in the vesicles of Oo-Mvi. Immunostaining showing GDF9 protein (Red, arrowheads) localized in the oocytes and the vesicles of Oo-Mvi (Green, RDX). $n = 29$ ovarian follicles. **b** Rhodamine-labeled GDF9 (R-GDF9) or BSA (R-BSA) was microinjected into living oocytes at GV stage ($n = 15$ per group), and the R-GDF9 intensity and distribution were detected at 15 or 30 min after injections (Supplementary Movies 4 and 5). The R-GDF9 spots surrounding the oocytes were pointed by arrowheads. **c**, **d** A time-dependent decrease of R-GDF9 intensity in oocytes (**c**) and increase of the number of R-GDF9 spots (**d**) were observed after injections. *P*-value: 0.0000014 (**c**), 0.00064 (**d**). **e** ER-tracker staining showed a cloudy ER distribution (top) in the oocyte cytoplasm, and the many ER bubbles (top, arrowheads) surrounding oocytes (Supplementary Movie 6), which were similar to the R-GDF9 distribution (**b**, arrowheads) and Oo-Mvi vesicles (bottom, arrowheads). **f** Quantification of the number of ER bubbles ($n = 17$) and Oo-Mvi vesicles ($n = 14$). *P*-value = 0.81. **g** Quantification of the fluorescence intensity of ER in oocyte cytoplasm and bubbles ($n = 20$). *P*-value = 0.00000009. **h** Relative mRNA levels of *Gdf9*, *Bmp15*, and *Fgf8* in Oo-Rdx$^{+/+}$ and Oo-Rdx$^{-/-}$ oocytes showing a decreased consumption of OSFs in the oocytes of Oo-Rdx$^{-/-}$ females. *P*-value: 0.000006 (*Gdf9*), 0.0001 (*Bmp15*), 0.0019 (*Fgf8*). **i** The proposed model of the function of Oo-Mvi in follicle development and female reproduction, showing that Oo-Mvi in the ZP determine the fate of individual follicles by regulating oocyte-GC communication. Data are presented as the mean ± SD with experiments performed in triplicate, and representative images are shown. Data were analyzed by two-tailed unpaired Student's *t*-test and \*\*\**P* < 0.001, \*\**P* < 0.01, and <sup>n.s.</sup>*P* ≥ 0.05. Scale bars: 5 μm (**a**); 20 μm (**b**, **e**).

cortex) or without (to detect the GC-TZPs) 0.1% Triton X-100 for 30 min to visualize the GC-TZPs by F-actin staining as previously reported[10,51].

To observe the cortical granules of oocytes, Oo-$Rdx^{+/+}$ and Oo-$Rdx^{-/-}$ oocytes at the GV stage (average diameter around 65 μm) were fixed in 4% PFA in PBS for 15 min and permeabilized in 0.5% Triton X-100 for 20 min at room temperature. Then, oocytes were blocked with 1% BSA (Sigma-Aldrich, V900933) in PBS for 1 h and incubated with LCA-FITC 2 h (1:100 dissolved in PBS, ThermoFisher Scientific, L32475, a gift from Dr. Bo Xiong in Nanjing Agricultural University, China) at room temperature[30]. After washing with PBS, the stained oocytes were imaged by an Andor Dragonfly spinning-disc confocal microscope as previously described indexes.

After the acquisition, Movies or single time-point images were processed by ImageJ (http://rsbweb.nih.gov/ij/) for projection of all z-stacks and merged color channels. To clearly highlight the structures of the Oo-Mvi, the mG (488 nm) channel was inverted to black and white by ImageJ software in Fig. 1. To show Oo-Mvi, R-GDF9, or ER of the oocyte, the rotary 3D Supplementary Movies were processed by Imaris (https://imaris.oxinst.com/) software (Supplementary Movies 1, 4, 5, and 6). To show the cellular behaviors of Oo-Mvi on oocytes, the time-lapse Supplementary Movies were processed by ImageJ software (Supplementary Movies 2 and 3).

The detailed protocol is also available in Protocol Exchange[52].

**High-resolution imaging of the ER in oocytes**. To observe the ER distribution of oocytes, denuded living oocytes at GV stage (average diameter around 65 μm) from fresh wild-type ovaries were incubated with ER-tracker Red (1:1000, Beyotime, C1041) for 30 min at 37 °C, 5% $CO_2$[32,33]. After washing with PBS, the stained oocytes were imaged by an Andor Dragonfly spinning-disc confocal microscope as previously described indexes. To the detection of ER fluorescent intensity of oocyte cytoplasm and bubbles, the single Z of mean ER fluorescent intensity of oocyte cytoplasm and bubbles were measured by ImageJ software.

**Immunofluorescence staining and observation**. The ovarian sections were deparaffinized, rehydrated, and subjected to high-temperature (95–98 °C) antigen retrieval for 16 min with 0.01% sodium citrate buffer (pH 6.0). The sections were then blocked with 10% normal donkey serum (Jackson ImmunoResearch, 017-000-121) for 60 min at room temperature and incubated with primary antibodies overnight at 4 °C. Subsequently, the sections were incubated with fluorophore-conjugated donkey secondary antibodies (1:200, Life Technologies) for 2 h at 37 °C. The homologous IgG (Goat, normal control, A7007, Beyotime) with the primary antibodies were used to be the negative control to guarantee the specificity of staining.

To detect ERM family or GDF9 protein expression in the ovary, we incubated sections with the following primary antibodies: the anti-RDX antibody (rabbit, 1:100, ab52495, Abcam), anti-p-ERM antibody (rabbit, 1:100, mAb#3726, Cell Signaling Technologies), anti-EZRIN antibody (rabbit, 1:100, ab4069, Abcam), anti-MOESIN antibody (rabbit, 1:100, ab52490, Abcam), and anti-GDF9 antibody (goat, 1:50, AF739, R&D). Cell proliferation of GCs was assessed by staining with the anti-proliferating cell nuclear antigen (PCNA) antibody (mouse, 1:100, Santa Cruz, sc-25280). The percentage of PCNA-positive GCs was quantified as the number of GCs with PCNA-positive signals divided by the total number of GCs in primary follicles and secondary follicles. Cell apoptosis was detected by TUNEL staining (In Situ Cell Death Detection Kit, Fluorescein, Roche, 11684795910). The percentage of TUNEL-positive follicles was quantified as the number of follicles with TUNEL-positive signal divided by the total number of follicles in the primary and secondary stages.

All general morphologies were examined and photographed by a Nikon Eclipse Ti digital fluorescence microscope.

For the high-resolution detections of GDF9, the co-localization of GDF9 with vesicles of Oo-Mvi was acquired as a z-stack model with an optical slice thickness of 0.3 μm after co-staining of GDF9 and RDX (to label Oo-Mvi), and confocal sections covered ~8 μm tissue thickness. Both the IgG and the ovarian sections from $Gdf9^{-/-}$ females were used as the negative control to guarantee the specificity of staining. The images were acquired by an Andor Dragonfly spinning-disc confocal microscope equipped with a ×100, 1.44 N.A. oil objective (Leica HC PL APO), a scientific complementary metal-oxide semiconductor (sCMOS) camera (Andor Zyla 4.2), and the 488-nm (anti-RDX, Oo-Mvi) and 568-nm (anti-GDF9 and IgG) lines of the Andor Integrated Laser Engine (ILE) system with a spinning-disc confocal scan head (Andor Dragonfly 500). In detail, images were acquired with laser 488-nm and laser 568-nm around 10–15%, exposure time 100–200 ms. Images were acquired by Fusion 2.1 software (https://andor.oxinst.com/products/dragonfly#fusion). After the acquisition, images were processed by ImageJ (http://rsbweb.nih.gov/ij/) for projection of all z-stacks and merged color channels.

**ZP collection and MS analysis**. To collect empty ZP from oocytes, C57BL/6 mice at PD23 with 5 IU of pregnant mare serum gonadotropin (PMSG; Sansheng Biological Technology) for 46 h and the ovaries were punctured to obtain cumulus-oocyte complexes. The cumulus cells were digested with 0.3% hyaluronidase medium (Merck, MR-051-F) to obtain denuded oocytes, which were then incubated with 10% high osmotic glucose solution to separate the ZP and the oocyte.

Under the microinjection system, the ZP was torn to release the oocyte, and then the empty ZP was collected. The protein (~5 μg) from ~3000 pieces of ZP was extracted by RIPA extraction buffer for a single protein MS assay, which was repeated three times.

Protein digestion was performed using the filter-aided sample preparation (FASP) method with modifications as previously described[53,54]. Nanospray electrospray (ESI)-MS was performed on a Thermo Q-Exactive high-resolution mass spectrometer (Thermo Scientific) with a 70,000 MS scan resolution, 17,500 MS/MS scan resolution, and top-10 MS/MS selection. Raw data from the mass spectrometer were preprocessed with Mascot Distiller 2.4 for peak picking. The resulting peak lists were searched against the Uniport Mouse database using the Mascot 2.5 search engine. Scaffold PTM was used to evaluate phosphorylation sites of the Mascot search results using the Ascore algorithm.

**Mouse fertility and ovulation analysis**. To detect mouse fertility, 5-week-old female Oo-$Rdx^{-/-}$ mice and control mice were continuously mated with 8-week-old C57BL/6 fertile males for 40 weeks. The numbers of pups and litters were recorded, from which fertility rates were determined.

For superovulation, Oo-$Rdx^{-/-}$ and control mice (PD23) were intraperitoneally injected with 5 IU of PMSG, (Sansheng Biological Technology, Ningbo, China) followed by 5 IU human chronic gonadotropin (hCG, Sansheng Biological Technology, Ningbo, China) 46 h later. After an additional 16 h, oocytes were collected from oviducts, and the numbers of oocytes for each animal were counted after digestion with 0.3% hyaluronidase (Merck, MR-051-F).

**In vitro culture of ovarian cortical pieces**. The in vitro attached culture of ovarian pieces was modified from the method in the previous study[55]. In details, ovarian cortical pieces from Oo-$Rdx^{+/+}$;mTmG and Oo-$Rdx^{-/-}$;mTmG ovaries at PD4 (oocyte diameter tracing experiment) were cut into approximately $2 × 1 ×$ $0.5 mm^3$ pieces and cultured in MEMα (Gibco, 32-571-036) supplemented with 10% FBS (Gibco, 10-099-141). After 4 days of adherent culture without changing the medium, the cortical pieces were attached firmly to the bottom of dishes for the experiments, and then the culture media was half-changed every day to maintain the development of tissues.

In the experiment of oocyte diameter tracing, the developmental dynamics of oocytes were measured by the increase of the diameter, which was labeled by mG in Oo-$Rdx^{+/+}$;mTmG and Oo-$Rdx^{-/-}$;mTmG ovaries. The diameter changes were recorded with a Nikon Eclipse Ti digital fluorescence microscope every other day for 7 continuous days. To ensure the detected oocytes were in a fast growth period, the oocytes with an initial diameter of around 16–35 μm were traced in this experiment. To compare the developmental dynamics of oocytes with different initial diameter, we normalized the actual diameters to the growth ratio to show the dynamics change of oocyte growth in different groups. Totally 151 oocytes from 27 cortical pieces in Oo-$Rdx^{+/+}$;mTmG and 250 oocytes from 21 cortical pieces in Oo-$Rdx^{-/-}$;mTmG ovaries were traced. In the GDF9-supplying experiment, Oo-$Rdx^{+/+}$ and Oo-$Rdx^{-/-}$ ovaries at PD21 (GDF9-supplying experiment) were cut to approximately $2 × 1 × 0.5 mm^3$ pieces and cultured in MEMα supplemented with 10% FBS, with or without GDF9 (500 ng/mL, R&D, 739-G9-010/CF) for 24 h, and the oocytes (average diameter around 65 μm at GV stages) were collected by tearing the ovaries with a syringe needle and digested with 0.3% hyaluronidase medium to be used for the detection of GC-TZP density.

**In vitro culture of ovarian follicles**. Follicles were separated by tearing PD21 ovaries and mechanical isolation with a syringe needle. The follicles with the diameter of around 130 μm (average diameter: 134.72 ± 13.73 μm in Oo-$Rdx^{+/+}$, $n = 64$ and 129.73 ± 20.21 μm in Oo-$Rdx^{-/-}$, $n = 77$) were used in the experiments. To monitor the growth of follicles, the isolated follicles were cultured in a Matrigel (BD, 354234) culture system to support a 3D development of follicles, which was modified from previously described[56–58]. Generally, the Matrigel was 3:1 diluted with pre-cooling MEMα-FBS-ITS medium-plus follicle-stimulating hormone (ovine FSH, 10 ng/mL, NHPP) on ice. Add 20 μL Matrigel/microdroplet in the 6 well culture plate and 6 microdroplets every well on ice keeping the gel in liquid. Put the 6 well culture plate at 37 °C for 20 min to trans Matrigel from the liquid phase to solid. Then, the isolated follicles were gently seeded into the gel with a mouse pipette, respectively. Adding the 2 μL liquid Matrigel to seal the hole and put the plate into the incubator at 37 °C for 10 min to make the gel sealed. After this step, the cultured follicles were completely surrounded by the gel. The density of seeding was around 1.6 per $cm^2$ to guarantee sufficient space for growth. Supplemented 1 mL MEMα-FBS-ITS medium-plus follicle-stimulating hormone (ovine FSH, 10 ng/mL, NHPP) into the culture plate. During culture, the media was half-changed every day, and the diameter of follicles was recorded every 24 h for 8 continuous days by a Nikon Eclipse Ti digital fluorescence microscope in a bright field channel.

The detailed protocol is also available in Protocol Exchange[59].

**Rhodamine-conjugated GDF9 and living oocyte injection**. Recombinant GDF9 protein (50 μg/mL, R&D, 739-G9-010/CF) or BSA (50 μg/mL, Sigma-Aldrich, V900933) as a control were labeled with rhodamine using the Pierce NHS-rhodamine antibody labeling kit (53031, ThermoFisher) according to the

manufacturer's protocol[60]. To collect the oocytes, the COCs were isolated by puncturing the antral follicles from C57BL/6 mice at PD23 after 46 h of PMSG treatment (5 IU, i.p. injection). After treated with 0.3% hyaluronidase (Merck, MR-051-F) in medium, denuded oocytes were performed microinjection, and approximately 10 pL of labeled protein or rhodamine solution were injected into one oocyte with a FemtoJet 4× electric microinjector (Eppendorf). After 15 or 30 min of injections, the oocyte was imaged under an Andor Dragonfly spinning-disc confocal microscope with the previously described indexes.

**Gene expression analysis**. To detect gene expression in mouse materials, the mRNA from different tissues or ovary compartments was extracted by TRIZOL Reagent (Thermo-Ambion, 15596018) according to the manufacturer's protocol. The quantity and quality of the total RNA were determined using a Nanodrop (Thermo Scientific). Reverse transcription (TAKARA, RR047Q) was performed using 1 µg total RNA per sample. QRT-PCR reactions were performed in 96-well plates (Applied Biosystems, 4316813) in 15 µL reaction volumes and analyzed by an Applied Biosystems 7500 Real-Time PCR System (Applied Biosystems, 4472908) using the following parameters: 10 min at 95 °C, followed by 40 cycles of 15 s at 95 °C and 1 min at 60 °C. Data were normalized to β-actin. The primer list is provided in Supplementary Table 1.

**Western blot**. To detect protein expression in mouse materials, total proteins were extracted using WIP Tissue and Cell lysis solution (BioChip, 110000) according to the manufacturer's protocol. Electrophoresis was performed with 50 µg total proteins separated by 10% SDS-PAGE and transferred to polyvinylidene fluoride membranes (Millipore, ISEQ00005). The membranes were incubated overnight at 4 °C with the appropriate primary antibodies listed below: anti-RDX (78 kDa, 1:500, ab52495, Abcam), anti-p-ERM (78 kDa, 1:500, mAb#3726, Cell Signaling Technologies). The appropriate secondary antibody (ZB-2301, ZB-2305 from ZSGB-BIO) was diluted 1:5000 in TBST (TBS plus 0.5% Tween 20). Tubulin (1:5000, Beyotime, AF5012) was used as an internal control. The membranes were visualized using the SuperSignal chemiluminescent detection system (Thermo Scientific, 32109). Original blots can be found in the Source data file.

**Statistical analysis**. All experiments were repeated at least three times using different mice. Data are presented as the mean ± standard deviation (SD) of each experiment. Data were analyzed by Student's $t$-test and were considered statistically significant at $P < 0.05$. P is indicated as follows: *($P < 0.05$), **($P < 0.01$), ***($P < 0.001$), and [n.s.](not significant, $P \geq 0.05$). Statistics and graphs were obtained using Prism 5 (GraphPad Software, La Jolla).

## Data availability

The mass spectrometry proteomics data have been deposited in the ProteomeXchange Consortium via the partner repository iProx[61] under the data set identifier IPX0002846000 for Fig. 2b and Supplementary Fig. 2 associated raw data (https://www.iprox.org/page/project.html?id=IPX0002846000). This is also found at the Proteome Xchange: see http://proteomecentral.proteomexchange.org/cgi/GetDataset?ID=PXD024931. All data are available from the corresponding author upon reasonable request. Source data are provided with this paper.

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

## Acknowledgements
We are grateful to Xin Li for drawing the illustrations in the manuscript and Dr. Lei Li, Dr. Bo Xiong, and Dr. Youqiang Su for kindly suggestions. This study was supported by the National Key Research and Development Program of China to Y.Z. and H.Z. (2018YFC1003800; 2018YFC1003700; 2017YFC1001100) and the National Natural Science Foundation of China (81873815 and 31571542).

## Author contributions
Y.Z., Y.W., G.X., and H.Z. designed the research; Y.Z., Y.W., X.F., S.Z., X.X., S.N., L.L., and Y.B. performed the experiments; Y.Z., Y.W., C.W., Z.L., and H.Z. analyzed the data; Y.Z., Y.W., and H.Z. wrote the paper. All authors have seen and approved the final version.

## Competing interests
The authors declare no competing interests.
