## [Peer Review File · Nature Communications]

Reviewers' Comments:

Reviewer #1:

Remarks to the Author:

The manuscript entitled "Unique oocyte-derived microvilli control female fertility through optimizing the ovarian 1 follicle selection in mice" investigate the role of oocyte derived microvilli. A series of experiments were conducted to investigate the role of these structures controlling OSF secretion. The manuscript is well designed and well written. However, some of the conclusions are too strong based on the data presented here. I raised some concerns related to the presented data. I am looking for an additional interpretation of the results.

Line 41: This sentence can be a little controversial "blocking the direct communications between oocytes" since transzonal projections (TZPs, look at Huge Clarke's work) derived from oocyte surrounding cells could mediate somatic-germline cell communication. Is it possible to have some GC derived TZPs formed before the ZP and new TZPs originated later on? If so, the ZP does not block the direct communication. Could you check this sentence?

Lines 63-65: could you please write the overall hypothesis of the work in addition to the text present in this paragraph?

Line 247: In this section authors suggest that the loss of Oo-Mv affects GCs proliferation due to its effects on OSF release, which was rescued by GDF9 supplementation. Authors did not show the levels of GDF9 in oocytes from control and Oo-RDX^{-/-} group. Could that be different? mRNA and protein results could answer this question.

Lines 308-313: Could you please explain the mechanism leading to blockage of mRNA consumption for the OSF. Did authors investigate the protein levels? Could the Oo-RDX^{-/-} affect mRNA translation as well as affecting OSF secretion due to reduced Oo-Mv? Also, in figure 6a you inject labelled GDF9 in WT oocytes to demonstrate the ability of Oo-Mv to secrete the labelled GDF9? However, only four spots were present in the picture. Do authors believe that the Oo-Mv are the main mechanism leading to OSF secretion? Why?

Lines 345-346: A direct correlation between lack of Oo-Mv and reproductive lifespan was not demonstrated. Instead authors demonstrated an association between the lack of Oo-Mv and shorter reproductive capacity. Could you please check it and explain the possible mechanisms driven shorter reproductive lifespan.

Line 361: What do authors mean with "vesicles heads". Could you explain it? Is there other cellular structure similar to that? Could you check the manuscript by Suzuki et al., 1998 in Journal of Reproduction and Development, Vol. 44, No. 4, 1998. Similar cellular structures are observed in that publication.

Lines 367-369: What evidences do authors have confirming the connection between clinical symptoms of premature ovarian insufficiency and the role of Oo-Mv?

Lines 369-376: Could you please revise these sentences. Conclusions within these sentences are strong and might not be fully answered by the results obtained in the manuscript.

Lines 146-149 in the M&M: I could not find the data related to this part of the material and methods? What happened when authors injected labelled GDF9 in RDX^{-/-} oocyte?

Also, regarding the experiment injecting labelled GDF9 in oocytes? If some cumulus cells are left

attached to the ZP, would observe the presence of labelled GDF9 in these cells? Does it happen in similar levels in WT and RDX^{-/-} mice?

Juliano da Silveira

Reviewer #2:

Remarks to the Author:

The development of a functional oocyte within the follicle requires communication between the germ cell and somatic (granulosa) cells that surround it. While the granulosa cells supply many factors to the oocyte, the oocyte releases members of the TGFbeta growth factor family, including growth-differentiation factor 9 (GDF9), which is required for normal proliferation and differentiation of the granulosa cells. It has long been known that transzonal projections that extend from the granulosa cells and contact the oocyte mediate much of this bi-directional communication. This manuscript from a well-established and highly respected team reports that microvilli extend from the oocyte, projecting towards the granulosa cells. The formation of these microvilli requires radixin, a member of the ERM family of membrane-associated proteins, and deletion of radixin from oocytes impairs oocyte development leading to severely reduced fertility. The authors conclude that these microvilli are the mechanism by which the oocyte releases GDF9; the lack of GDF9 in the mutants thus underlies the observed phenotype. While it has been known that oocytes produce microvilli, their function has so far not been uncovered and this manuscript could fill this important knowledge gap.

The first half (Figs. 1-3) is very strong. The authors unambiguously demonstrate that the oocyte microvilli exist (Fig. 1), that the oocyte expresses radixin with cortical localization of a phosphorylated form (Fig. 2), and that oocyte-specific knockout of radixin causes loss of the microvilli and reduces fertility (Fig. 3). A point the authors need to address is that a previous report of a global radixin knockout indicated that the mutant mice were fully fertile up to at least one year of age (Nat Gen 31, 320).

The second half of the manuscript (Figs. 4-6), which addresses how loss of radixin impairs oocyte development, is less convincing. First, the ERM proteins are believed to play a wide range of roles in organizing and/or enabling the function of the cell cortex (eg, Nat Rev Mol Cell Biol 11, 276). Deletion of radixin is likely to disrupt many cortex-dependent activities in the oocyte in addition to production of the microvilli. Could these not be responsible for the reduced fertility? This issue is not addressed in the manuscript. The authors need to provide a convincing and experimentally supported rationale for believing that, apart from the absence of microvilli, cortical structure and function in the mutant oocytes is otherwise normal.

Second, no evidence is provided in the manuscript that GDF9 is present in the bulbs at the end of the oocyte microvilli or that GDF9 secretion by the mutant oocytes is actually impaired. This is essential, as it underpins the model. Co-localization should be possible using fluorescence as in Fig. 6b (which does not show co-localization) and previous work by the Eppig and Matzuk labs has identified consequences or markers of a lack of GDF9 that could be assessed in this model.

Third, the phenotype of this knockout does not match well with that described for the Gdf9 knockout. For example, the loss of follicles here (Fig. 4b) has not been reported in the Gdf9 ko, and the reduced oocyte growth here (Fig. 4d) is the opposite of the Gdf9 ko phenotype. Apart from reduced fertility and absence of TZPs, it is difficult to see how the two phenotypes resemble each other.

An additional concern is that little methodological detail is provided with respect to many of the experiments shown in Figures 4-6. The information supplied is not sufficient to permit the experiments to be reproduced. A few examples:

- How were follicles identified as atretic (4a)?
- What are the stained structures in 4c? The size suggests oocytes, but the morphology suggests follicles.
- Why not express actual oocyte diameter in 4d?
- How were follicles and oocytes recovered from the cultured ovarian fragments?
- References cited (36, 50, 51) for culture of follicles at different stages do not describe extended culture of post-natal follicles. More detail is needed.
- Fig 5g shows that the number of TZPs tripled after a 24-hr culture with GDF9. However, the number of oocytes examined is very small, ranging from 7-11. How many independent biological replicates were performed – should be a minimum of three.

Finally, the manuscript should be edited for clarity and accuracy. Again, a few examples:

Line 59: How is it known that the zona pellucida reduces the efficiency of OSFs?

Line 103: The observation that the head vesicles break down is interesting, but does not demonstrate a function.

Line 134: The statement that Ezrin and Moesin mRNA were 'hardly detected' in oocytes is not an accurate representation of the data shown in the Figure.

Line 344: What is the direct evidence that the Oo-Mv regulate OSF release?

Reviewer #3:

Remarks to the Author:

The authors have performed an extensive set of studies that have revealed a type of microvillus present on mouse oocytes that has a mushroom-shaped body at the distal end and that extends through most of the zona pellucida. They show that ablation of radixin in oocytes caused the loss of these microvilli. They also present evidence that the oocyte-secreted factor GDF9 was partly localized in these microvilli in the tip. Deleting radixin decreased the amount of oocyte-secreted factors and also decreased the numbers of transzonal processes from the granulosa cells. Females with radixin knocked down in their oocytes were subfertile and became infertile at an earlier age.

This is a very interesting and potentially very significant set of findings. The identification of a new anatomical structure in mammalian oocytes is particularly significant.

1) The main weakness is with the evidence connecting the effects of deleting oocyte radixin specifically with the loss of these types of microvilli. None of the experiments conclusively show that specifically ablating the long microvilli with the mushroom-shaped terminus is responsible for the reported effects on cumulus transzonal processes or female fertility.

Radixin is unlikely to be specific only to these long microvilli in oocytes. Indeed, in figure 2 e-h, it is clear that radixin protein is primarily localized in the cytoplasm (likely being trafficked to the membrane) and in a thick band at the cortex as well as in the long microvilli, with most in the cortical band. Phosphorylated radixin is localized primarily in the cortical band along with the long microvilli. When knocked out, all of this radixin disappears as expected. The identity of the cortical band is unclear. It is very well known that oocytes are densely covered with microvilli, which are presumably present in a greater number than the structures that are the focus of this report. Is the cortical band those microvilli? Or is in the subcortical actin band in oocytes? Perhaps both? The overexposure of this region to highlight the long microvilli makes it impossible to tell. Either was, the subcortical actin and/or the actin in the more numerous short microvilli could easily be disrupted by the loss of radixin and could account for the effects on cumulus cell transzonal processes and female fertility. In summary, much stronger evidence is needed before it is claimed that these phenotypes are specifically due to the loss of the long microvilli. Other possibilities could be an effect on secretion, particularly of zona pellucida proteins or oocyte-derived factors at the oocyte surface, loss of small microvilli, a defect in zona pellucida biosynthesis, mislocalization of

the egg's cortical granules, meiotic spindle mislocalization, or other effects.

2) The authors should specify the stage of oocyte shown in each figure. Some oocytes are at various stages of growth while some are fully-grown. The oocyte stage is only clearly indicated in fig 1d.

3) The evidence showing that GDF9 is localized in the tips of the long microvilli is weak, depending on similar round structures being located in the same region of the zona pellucida and being of similar diameters. Colocalization with a marker for these long microvilli should be shown (e.g., radixin or mG).

4) A more minor point is that the boundaries of the oocyte surface, the zona pellucida, and the innermost granulosa cells are not clear in the fluorescence micrographs. These should be indicated by labeling in the figures.

General additional results to the reviewers:

Oocytes control the OSF release by regulating the formation of Oo-Mv

To clarify the functional role of Oo-Mv in regulating the release of OSFs, we expanded the data of Rhodamine-labeled GDF9 (R-GDF9) injections by performing a time-dependent detection (Fig. R1a) after microinjection. As a result, we found a clearly reduce of Rho-fluorescent intensity with the time increase after injection in oocytes, showing a fast release of the R-GDF9 from oocytes (Fig. R1a and b). With the decrease of R-GDF9 signal, a time-dependent increase of the R-GDF9 spot number (Fig. R1a, arrowheads and Supplementary Video 4-5) was observed surrounding the injected oocytes. As we showed in the Fig. R1c, the average number of R-GDF9 spots was increased from 8 ± 4 at ~ 15 mins to 25 ± 9 at ~ 30 mins after injection. These data clearly clarified that labeled GDF9 was released from oocyte through Oo-Mv. (The related information was also added in results of main text from line 308 to 313 and Fig.6b-d in the revised manuscript.)

[Redacted]

[Redacted]

Finally, the existence of GDF9 in the bubbles (or vesicles) of Oo-Mv was detected through high-resolution imaging by co-staining of RDX and GDF9 in the adult ovaries. As we showed in Fig. R1f, the signal of GDF9 was clearly observed in the bubble of Oo-Mv in the ZP which was labeled by the Radixin staining. This result clearly showed that GDF9 distributed in the tip of Oo-Mv, indicating OSFs were enriched in the vesicles of Oo-Mv to be released. Taking together, these data supplied directly experimental evidence showing that the oocyte controls the release of OSF through regulating the formation of Oo-Mv. (The related information was also added in results of main text from line 293 to 297 and Fig. 6a in the revised manuscript.)

[Redacted]

[Redacted]

Deleting *Rdx* mainly affect the formation of Oo-Mv on oocytes

To validate the effects of *Rdx* deletion in oocytes, we first detected the construction of oocyte cortex in Oo-*Rdx*^{-/-} females followed standard approach by labeling the actin with Alex488-phalloidin in oocytes^{1,2}. After staining, although less TZPs were found in the ZP of oocytes from Oo-*Rdx*^{-/-} females compared to Oo-*Rdx*^{+/+} controls, an identical distribution and thickness of the oocyte cortex were observed in Oo-*Rdx*^{+/+} and Oo-*Rdx*^{-/-} oocytes (Fig. R2a-b). This result showed that the deletion of *Rdx* had no effect on the formation of oocyte cortex. (The related information was also added in results of main text from line 173 to 176 and Extended Data Fig. 4 in the revised manuscript.)

Furthermore, we collected the superovulated oocytes from Oo-*Rdx*^{-/-} and Oo-*Rdx*^{+/+} females and performed the *in vitro* fertilization. Although less oocytes were collected from mutant females, all of these mutant oocytes were fertilized successful and formed the zygotes. Moreover, a comparable developmental capability of zygotes was found in Oo-*Rdx*^{+/+} and Oo-*Rdx*^{-/-} females till the blastocyst stage (Fig. R2c-d). These results showed that the oocytes from Oo-*Rdx*^{-/-} females were fertile if they could pass the process of follicle selection. Therefore, we concluded that deletion of *Rdx* had no dramatic effects on the oocyte survival except the failure of Oo-Mv formation. With these data, we believed that deleting *Rdx* caused the Oo-Mv failure were the major if not the only phenotypes in the oocytes.

Fig. R2. Deleting *Rdx* mainly affect the formation of Oo-Mv on oocytes. a-b, A normal formation of the oocyte cortex in Oo-*Rdx*^{-/-} female. Actin staining showing an abnormal formation of GC-TZPs construction in Oo-*Rdx*^{-/-} oocytes, but an identical oocyte cortex was found in Oo-*Rdx*^{-/-} (n = 17) and Oo-*Rdx*^{+/+} (n = 18) oocytes. **c-d,** The *in vitro* fertilization of Oo-*Rdx*^{-/-} and Oo-*Rdx*^{+/+} oocytes. The superovulated oocytes of Oo-*Rdx*^{-/-} females and Oo-*Rdx*^{+/+} females were performed *in vitro* fertilization. No significantly difference in the rate of zygote formation and embryonic development was found. Data are presented as the mean ± SD with experiments performed in triplicate, and representative images are shown. Data were analyzed by 2-tailed unpaired Student's *t*-test and n.s. P ≥ 0.05. Scale bars in (a): 20 μm, in (c): 100 μm

Reviewer #1 (Remarks to the Author):

The manuscript entitled “Unique oocyte-derived microvilli control female fertility through optimizing the ovarian follicle selection in mice” investigate the role of oocyte derived microvilli. A series of experiments were conducted to investigate the role of these structures controlling OSF secretion. The manuscript is well designed and well written. However, some of the conclusions are too strong based on the data presented here. I raised some concerns related to the presented data. I am looking for an additional interpretation of the results.

Line 41: This sentence can be a little controversial “blocking the direct communications between oocytes” since transzonal projections (TZPs, look at Huge Clarke’s work) derived from oocyte surrounding cells could mediate somatic-germline cell communication. Is it possible to have some GC derived TZPs formed before the ZP and new TZPs originated later on? If so, the ZP does not block the direct communication. Could you check this sentence?

Author response: Thanks for the valuable suggestion. The elegant study from Dr. Clarke presented the beautiful structure of TZPs, which connect the GCs with oocytes². We also realized that “blocking the direct communications between oocytes” is not accurate. Before the ZP formation, the cell membrane of dormant oocyte and surrounding pre-granulosa cells are closely attached. With the formation of ZP, this direct attachment of cell membrane is separated between oocyte and granulosa cells, and the TZPs are formed to construct the communications between oocytes and GCs. Therefore, we corrected the sentence to “blocking the direct cellular membrane attachment between oocytes and GCs” in the revised manuscript line 38.

Lines 63-65: could you please write the overall hypothesis of the work in addition to the text present in this paragraph?

Author response: Thanks for the valuable suggestion. We have added the overall hypothesis in the first sentence of this paragraph (line 60 to 62): ‘In the current study, we reported oocyte derived unique microvilli system mediated an orderly and timely OSF release which contributed to the fine regulation of ovarian follicle development and female reproductive lifespan maintenance.’

Line 247: In this section authors suggest that the loss of Oo-Mv affects GCs proliferation due to its effects on OSF release, which was rescued by GDF9 supplementation. Authors did not show the levels of GDF9 in oocytes from control and Oo-RDX-/- group. Could that be different? mRNA and protein results could answer this question.

Author response: Thanks for the valuable suggestion. As we showed in fig. 6g of main manuscript, the mRNA levels of OSFs including *Gdf9* were significantly elevated in

Oo-*Rdx*^{-/-} group. This is one of several evidence showing that Oo-Mv is functional to control the OSF release. *[redacted]*

[Redacted]

[Redacted]

[Redacted]

Lines 308-313: Could you please explain the mechanism leading to blockage of mRNA consumption for the OSF. Did authors investigate the protein levels? Could the Oo-RDX^{-/-} affect mRNA translation as well as affecting OSF secretion due to reduced Oo-Mv? Also, in figure 6a you inject labelled GDF9 in WT oocytes to demonstrate the ability of Oo-Mv to secrete the labelled GDF9? However, only four spots were present in the picture. Do authors believe that the Oo-Mv are the main mechanism leading to

OSF secretion? Why?

Author response: Thanks for the valuable suggestion.

[Redacted]

This result indicated that the deletion of *Rdx* reduced the release of GDF9 but had no effect on the translation of *Gdf9* mRNA. Since the accumulation of mRNA in oocytes is mainly occurred at early growing stage to establish a mRNA reserve for further consumption³, we believed that GDF9 was translated in the cytoplasm of Oo-*Rdx*^{-/-} oocytes but could not release to extracellular space efficiently, which caused a reduced consumption of *Gdf9* mRNA in the Oo-*Rdx*^{-/-} oocytes (Fig. 6g in main manuscript).

For the Rho-labeled GDF9 injecting experiment, we are sorry for the misleading of the data. Although we highlighted 4 spots in the light section of Fig. 6b in main manuscript, there were lots of spots on the surface of injected oocyte under whole-mount scanning. We have supplied 3D reconstructed oocytes on the Supplementary Video 4-5. In those 3D movies, it is easy to observe the spots surrounding the surface of oocytes.

By performing the further experiments of R-GDF9 microinjection, we found that there was a time-dependent increase of the spot number on oocytes after R-GDF9 injection (Fig. R1-2a). Meanwhile, a clearly time-dependent reduce of Rho-fluorescent intensity was observed in the oocytes after injection, indicating a fast release of the GDF9 from oocytes (Fig. R1-2b). As we showed in the Fig. R1-2c, the average number of R-GDF9 spots was increased from 8 ± 4 at ~15 mins to 25 ± 9 at ~30 mins after injection. These data confirmed that labeled GDF9 was released from oocyte through Oo-Mv.

Fig. R1-2. OSF was fast released from oocyte through Oo-Mv. **a**, Rhodamine labeled GDF9 (R-GDF9) was microinjected into living oocytes at GV stage ($n = 15$ per group), and the R-GDF9 intensity and distribution were detected at 15 or 30 mins after injections. **b-c**, A time-dependent decrease of R-GDF9 intensity in oocytes (**b**) and an increase of the number of R-GDF9 spots (**c**) were observed after injections. Data are presented as the mean \pm SD with experiments performed in triplicate, and representative images are shown. Data were analyzed by 2-tailed unpaired Student's *t*-test and *** $P < 0.001$. Scale bars: 20 μ m.

Lines 345-346: A direct correlation between lack of Oo-Mv and reproductive lifespan was not demonstrated. Instead authors demonstrated an association between the lack of Oo-Mv and shorter reproductive capacity. Could you please check it and explain the possible mechanisms driven shorter reproductive lifespan.

Author response: Thanks for the valuable suggestion, and we believed this was an interested point in our finding. Actually, we had expected an infertile phenotype when we observed the smooth oocyte from Oo-*Rdx*^{-/-} females under the microscope. But the fertility check results showed that females were fertile with a reduced pup number after mating. We therefore extended the period of fertility check and found a dramatically reduced reproductive lifespan.

For the underlying mechanisms of shortened reproductive lifespan, we believed that the Oo-Mv deficiency led to a decreased efficiency of OSFs release rather than a complete block of OSFs function. We found an increased loss of follicles in the Oo-*Rdx*^{-/-} ovaries, but part of follicles were still able to develop to antral stage. This finding was same as the reported phenotype of *Bmp15*^{-/-}*Gdf9*^{+/-} females from Dr. Matzuk's lab⁴. In *Bmp15*^{-/-}*Gdf9*^{+/-} females, the dramatic reduce of OSFs release caused a dramatically loss of follicles in the ovaries of mutant females. Moreover, those mutant females had a dramatic decreased fertility and failed to produce any offspring after 6-month of mating, which is consistent with the phenotypes of Oo-*Rdx*^{-/-}. Therefore, we concluded that deletion of *Rdx* mainly affected the formation of Oo-Mv and led to a low efficiency of OSF release, which finally caused a reduced reproductive capacity in females.

Line 361: What do authors mean with "vesicles heads". Could you explain it? Is there other cellular structure similar to that? Could you check the manuscript by Suzuki et al., 1998 in Journal of Reproduction and Development, Vol. 44, No. 4, 1998. Similar cellular structures are observed in that publication.

Author response: Thanks for the valuable suggestion. We defined the vesicles-like expanded membrane in the tip of Mv in our manuscript (Line 91). In the pioneer study from Suzuki et al., they reported the similar structures on the membrane of cleaved bovine eggs at 2 or 3 cells stage⁵. The interested findings from Suzuki and us, indicated that this kind of Mv might be a conserved structure during the oocyte and early embryo development. We have added this discussion in the line 398 to 399 of revised manuscript.

Lines 367-369: What evidences do authors have confirming the connection between clinical symptoms of premature ovarian insufficiency and the role of Oo-Mv?

Author response: Thanks for the valuable suggestion. Generally, the females with premature loss of ovarian follicles, decreased reproductive lifespan and infertile were defined as premature ovarian insufficiency in the mouse model study^{6,7,8}. This is one of the reasons that we claimed a POI phenotype in our study. The other reason is that the

mutation of *RDX* was found in part of POI patients in the GWAS data of POI patients from our collaborator in the Shandong University⁹. Therefore, we believed that the loss of Oo-Mv in *Rdx* mutation should be a potential etiology of human POI. We are sorry that the detail data cannot be shared here since the study is ongoing in the lab of our collaborator.

Lines 369-376: Could you please revise these sentences. Conclusions within these sentences are strong and might not be fully answered by the results obtained in the manuscript.

Author response: Thanks for the valuable suggestion. We have corrected the conclusion in line 393 to 402. ‘With our experimental data, we proposed that the role of Oo-Mv is an optimized system to select the qualified follicles, especially the early growing follicles. Meanwhile, although our results showed that OSFs are released through Oo-Mv, we also hypothesized that the release of OSFs does not completely rely on Oo-Mv since folliculogenesis never passed the secondary stage in *Gdf9*-KO females¹⁰. Previous study reported that oocytes also form membrane fusion with GCs to interact the secretion of GDF9, those observations further supported our hypothesis that Oo-Mv play a regulating role in controlling folliculogenesis¹⁸. Therefore, we believed that the major function of Oo-Mv is to make the release of OSFs timely and orderly, guaranteeing a precise and efficient regulation to determine the developmental fate of ovarian follicles.’

*Lines 146-149 in the M&M: I could not find the data related to this part of the material and methods? What happened when authors injected labelled GDF9 in *RDX*^{-/-} oocyte?*

Author response: Thanks for the valuable suggestion. Sorry we didn’t include the data of R-GDF9 injection into the Oo-*Rdx*^{-/-} oocytes in the previous manuscript. In consistent to the phenotype of no Oo-Mv formation on Oo-*Rdx*^{-/-} oocytes, there was no spots on the surface of Oo-*Rdx*^{-/-} oocytes after R-GDF9 injection (Fig. R1-3).

Fig. R1-3. Visualizing the enrichment of GDF9 in the Oo-Mv. Rhodamine labeled GDF9 (R-GDF9) was microinjected into living oocytes of Oo-*Rdx*^{+/+} and Oo-*Rdx*^{-/-}. R-GDF9 fluorescent bubbles (arrowheads) in the ZP were observed in R-GDF9-injected Oo-*Rdx*^{+/+} oocytes (up), but not in Oo-*Rdx*^{-/-} oocytes (down). Scale bars: 25 μ m.

Also, regarding the experiment injecting labelled GDF9 in oocytes? If some cumulus cells are left attached to the ZP, would observe the presence of labelled GDF9 in these cells? Does it happen in similar levels in WT and RDX^{-/-} mice?

Author response: Thanks for the valuable suggestion. Indeed, cumulus cells (CCs) were attached in a few of oocytes in the R-GDF9 experiments. As we showed in the Fig. R1-4, the Rho fluorescent signal was clearly observed in the CCs of Oo-Rdx^{+/+} oocytes, which was in sharp contrast to a background level of red fluorescence in the CCs of Oo-Rdx^{-/-} group. This result further clarified that a failure of OSF release in the Oo-Rdx^{-/-} oocytes.

Fig. R1-4. A failure of GDF9 release in Oo-Rdx^{-/-} female. The R-GDF9 was microinjected into living oocytes with cumulus cells (CCs). The Rho fluorescent signal was observed on CCs of Oo-Rdx^{+/+} oocytes, but not on the CCs from Oo-Rdx^{-/-} ovaries. Scale bars: 20 μ m.

Juliano da Silveira

Reviewer #2 (Remarks to the Author):

The development of a functional oocyte within the follicle requires communication between the germ cell and somatic (granulosa) cells that surround it. While the granulosa cells supply many factors to the oocyte, the oocyte releases members of the TGFbeta growth factor family, including growth-differentiation factor 9 (GDF9), which is required for normal proliferation and differentiation of the granulosa cells. It has long been known that transzonal projections that extend from the granulosa cells and contact the oocyte mediate much of this bi-directional communication. This manuscript from a well-established and highly respected team reports that microvilli extend from the oocyte, projecting towards the granulosa cells. The formation of these microvilli requires radixin, a member of the ERM family of membrane-associated proteins, and deletion of radixin from oocytes impairs oocyte development leading to severely reduced fertility. The authors conclude that these microvilli are the mechanism by which the oocyte releases GDF9; the lack of GDF9 in the mutants thus underlies the observed phenotype. While it has been known that oocytes produce microvilli, their function has so far not been uncovered and this manuscript could fill this important knowledge gap.

The first half (Figs. 1-3) is very strong. The authors unambiguously demonstrate that the oocyte microvilli exist (Fig. 1), that the oocyte expresses radixin with cortical localization of a phosphorylated form (Fig. 2), and that oocyte-specific knockout of radixin causes loss of the microvilli and reduces fertility (Fig. 3). A point the authors need to address is that a previous report of a global radixin knockout indicated that the mutant mice were fully fertile up to at least one year of age (Nat Gen 31, 320).

Author response: Thanks for the valuable suggestion. There are two potential reasons for the difference of the phenotypes in our study and previous study from Dr. Tsukita lab¹¹. The first one is the difference of deleting strategies in the study. The exon 4 and 5 of *Rdx* were deleted in our study, which is different with the exon 3 in previous report^{11,12}. It was well-known that the difference of the deleting strategy in same gene might lead to a variety of phenotype^{13,14}. The secondary reason might be the background of the mice. In our study, we backcrossed the mice to C57BL/6 inbred background for the study. From the M&M of Dr. Tsukita's manuscript, we didn't find the information about the backcross of the *Rdx* deficient mouse, therefore a mixed background of animals could be used in their study. Generally, the mild phenotype might not be appeared in the mixed background mice¹⁴. We guessed these should be the reasons that different phenotypes were found in our study and previous report.

The second half of the manuscript (Figs. 4-6), which addresses how loss of radixin impairs oocyte development, is less convincing. First, the ERM proteins are believed to play a wide range of roles in organizing and/or enabling the function of the cell cortex (eg, Nat Rev Mol Cell Biol 11, 276). Deletion of radixin is likely to disrupt many cortex-dependent activities in the oocyte in addition to production of the

microvilli. Could these not be responsible for the reduced fertility? This issue is not addressed in the manuscript. The authors need to provide a convincing and experimentally supported rationale for believing that, apart from the absence of microvilli, cortical structure and function in the mutant oocytes is otherwise normal.

Author response: Thanks for the valuable suggestion. Indeed, loss of Oo-Mv might not be the only defect of *Rdx* deletion in oocytes. To validate the effects of *Rdx* deletion in oocytes, we first detected the construction of oocyte cortex in Oo-*Rdx*^{-/-} females followed standard approach to label the actin with Alex488-phalloidin^{1,2}. After staining, although less TZPs were found in the ZP of oocytes from Oo-*Rdx*^{-/-} females compared to Oo-*Rdx*^{+/+} controls, an identical oocyte cortex was observed in Oo-*Rdx*^{+/+} and Oo-*Rdx*^{-/-} oocytes (Fig. R2-1. a-b). This result clearly showed that the deletion of *Rdx* had no significantly effect on the formation of oocyte cortex.

Furthermore, we collected the superovulated oocytes from Oo-*Rdx*^{-/-} females and performed the *in vitro* fertilization. Although less oocytes were collected from mutant females, all of these mutant oocytes were fertilized successful and formed the zygotes. Moreover, a comparable developmental capability of zygotes was found in Oo-*Rdx*^{+/+} and Oo-*Rdx*^{-/-} females till the blastocyst stage (Fig. R2-1. c-d). These results showed that the oocytes from Oo-*Rdx*^{-/-} females were fertile if they could pass the process of follicle selection. Therefore, we concluded that deletion of *Rdx* had no dramatic effects on the oocyte survival except the failure of Oo-Mv formation. With these data, we believed that deleting *Rdx* caused the Oo-Mv failure were the major if not the only phenotypes in the oocytes.

Fig. R2-1. Deleting *Rdx* mainly affect the formation of Oo-Mv on oocytes. a-b, A normal formation of the oocyte cortex in Oo-*Rdx*^{-/-} female. Actin staining showing an abnormal formation of GC-TZPs construction in Oo-*Rdx*^{-/-} oocytes, but an identical oocyte cortex was found in Oo-*Rdx*^{-/-} (n = 17) and Oo-*Rdx*^{+/+} (n = 18) oocytes. c-d, The *in vitro* fertilization of Oo-*Rdx*^{-/-} and Oo-*Rdx*^{+/+} oocytes. The superovulated oocytes of Oo-*Rdx*^{-/-} females and Oo-*Rdx*^{+/+} females were performed *in vitro* fertilization. No significantly difference in the rate of zygote formation and embryonic development was found. Data are presented as the mean ± SD with experiments performed in triplicate, and representative images are shown. Data were analyzed by 2-tailed unpaired Student's *t*-test and n.s. P ≥ 0.05. Scale bars in (a): 20 μm, in (c): 100 μm .

Second, no evidence is provided in the manuscript that GDF9 is present in the bulbs at the end of the oocyte microvilli or that GDF9 secretion by the mutant oocytes is actually impaired. This is essential, as it underpins the model. Co-localization should be possible using fluorescence as in Fig. 6b (which does not show co-localization) and previous work by the Eppig and Matzuk labs has identified consequences or markers of a lack of GDF9 that could be assessed in this model.

Author response: Thanks for the valuable suggestion. The existence of GDF9 in the bubbles of Oo-Mv was detected through high-resolution imaging by co-staining of RDX and GDF9 in the adult ovaries. As we showed in Fig. R2-2, the signal of GDF9 was clearly observed in the bubble of Oo-Mv in the ZP which was labeled by the Radixin staining. This result clearly showed that GDF9 distributed in the tip of Oo-Mv, indicating OSFs were enriched in the vesicles of Oo-Mv to be released.

Fig. R2-2. GDF9 was localized in the vesicles of Oo-Mv. Immunostaining showing GDF9 protein (Red) localized in the oocytes and the vesicles of Oo-Mv (Green, labeled by RDX antibody). Scale bars: 5 μm.

Moreover, to confirm the ovarian phenotype derived from the lack of GDF9, we selected a panel of reported downstream genes including *Has2*, *Ptgs2*, *Ptx3* and *Tnfnip6* which were reported to be regulated by GDF9 in cumulus-oocyte complexes¹⁵. We found a significantly decreased expression of *Ptgs2* and *Tnfnip6* in *Oo-Rdx*^{-/-} females (Fig. R2-3), indicating deletion of *Rdx* led to a GDF9 related deficiency of ovaries. (The related information was also added in results of main text from line 274 to 277 and Extended Data Fig. 7 in the revised manuscript.)

Fig. R2-3. The mRNA expressions of *Ptgs2* and *Tnfnip6* were decreased in *Oo-Rdx*^{-/-} females. Decreased mRNA levels of *Ptgs2* and *Tnfnip6* in *Oo-Rdx*^{-/-} cumulus-oocyte complexes (COCs) compared to the levels in *Oo-Rdx*^{+/+} COCs confirmed the deletion of *Rdx* led to a GDF9 related deficiency of ovaries. Data are presented as the mean \pm SD with experiments in triplicate. Data were analyzed by 2-tailed unpaired Student's *t*-test and ** P < 0.01.

Third, the phenotype of this knockout does not match well with that described for the Gdf9 knockout. For example, the loss of follicles here (Fig. 4b) has not been reported in the Gdf9 ko, and the reduced oocyte growth here (Fig. 4d) is the opposite of the Gdf9 ko phenotype. Apart from reduced fertility and absence of TZPs, it is difficult to see how the two phenotypes resemble each other.

Author response: Thanks for the valuable suggestion. In the pioneering *Gdf9* KO study from Dr. Matzuk in 1996, they reported that the follicle development cannot pass the primary stage, but there was no age-related follicle counting results in the manuscript¹⁰. However, the follow-up study from Dr. Matzuk's lab reported a dramatically loss of follicles in the ovaries of *Bmp15*^{-/-}*Gdf9*^{+/-} females⁴, and those females represented a decreased fertility in young adult and failed to produce any offspring after 6-month of mating, which is consistent with the phenotypes of *Oo-Rdx*^{-/-}.

With reviewer's enlightening suggestion, we also detected the formation of Oo-Mv on oocytes of *Gdf9*-KO ovaries which were generated by deleting exon 2 of *Gdf9* in our lab previously. These females represented identical phenotypes of *Gdf9*-KO females from Dr. Matzuk lab (Fig. R2-4. a)¹⁰. Interestingly, we found a significantly increased

density of Oo-Mv on the growing oocytes of *Gdf9-KO* females (Fig. R2-4. b-c). This result indicated that loss of GDF9 led to a compensatory increase of Oo-Mv number on the oocytes, which confirmed the function of Oo-Mv in regulating the release of OSFs.

Fig. R2-4. The density of Oo-Mv increased on the growing oocytes of *Gdf9-KO* females.

a, Histological analysis of the ovarian morphology in *Gdf9-KO* female at PD35, showing a decrease in ovarian size and developmental retardation of follicles in the mutant ovaries. **b-c**, RDX immunostaining to label Oo-Mv showing that the density of Oo-Mv was significantly increased on the growing oocytes of *Gdf9-KO* follicles (n = 31) comparing to control (n = 26). Data are presented as the mean ± SD with experiments performed in triplicate, and representative images are shown. Data were analyzed by 2-tailed unpaired Student's *t*-test and *** P < 0.001. Scale bars in **(a)**: 200μm, in **(b)**: 10 μm.

*An additional concern is that little methodological detail is provided with respect to many of the experiments shown in Figures 4-6. The information supplied is not sufficient to permit the experiments to be reproduced. A few examples:
-How were follicles identified as atretic (4a)?*

Author response: Thanks for the comments. The follicles with clearly debris of oocytes were defined to atretic follicles in Figure 4a (the enlarged figure as shown Fig. R2-5). (The related information was also added in results of Extended Data Fig. 6b in the revised manuscript.)

Fig. R2-5. The atretic follicles with debris of oocytes in *Oo-Rdx*^{-/-} ovaries at 2M. The debris of oocytes in atretic follicles were pointed by arrows. Scale bars: 50 μm.

-What are the stained structures in 4c? The size suggests oocytes, but the morphology suggests follicles.

Author response: Thanks for the comments. In this figure, we cultured ovarian cortical pieces which were isolated from *Gdf9-Cre;mTmG* females at PD4. After 4 days of culture, the cortical pieces were attached on the bottom of dishes, and the oocytes were identified by the GFP outline to measure the diameters (Fig. R2-6). To highlight the changes of oocyte diameter, we chose the representative oocytes which the growth was traced under microscope to show in the figure 4c of main text.

Fig. R2-6. The fluorescent image showed growing oocytes (Green) in the ovarian cortical pieces of PD4 *Gdf9-Cre;mTmG* females after 4 days culture. Scale bars: 50 μ m.

-Why not express actual oocyte diameter in 4d?

Author response: Thanks for the comments. To ensure the detected oocytes were in a fast growth period (less than 35 μ m), we selected the oocytes with an initial diameter around 16 to 35 μ m in this experiment. Since the variation of the initial diameter of different oocytes, we normalized the actual diameters to the growth ratio to show the dynamics change of oocyte growth in different groups. We have expanded the method and supplied methodological detail in the revised manuscript (line 518 to 521).

-How were follicles and oocytes recovered from the cultured ovarian fragments?

Author response: We are sorry for the unclear description of this details in method. We have expanded the method and supplied methodological detail in the revised manuscript (line 525 to 526 and line 528 to 530).

-References cited (36, 50, 51) for culture of follicles at different stages do not describe extended culture of post-natal follicles. More detail is needed.

Author response: We are sorry for the miss of the related references. The method of 3D culture of follicles in Matrigel was as described previous study^{16, 17}. We have modified the references cited (49, 53, 54) at Line 530, and the detail method was supplied in line 528 to 530 of the revised manuscript.

-Fig 5g shows that the number of TZPs tripled after a 24-hr culture with GDF9. However, the number of oocytes examined is very small, ranging from 7-11. How many independent biological replicates were performed – should be a minimum of three.

Author response: Thanks for the valuable suggestion. In the initial manuscript, all experiments were performed more than triplicate independent biological replicates. Since the collection and staining of oocytes from cultured ovarian pieces was not easy, the total sample number was not satisfactory. In the revised manuscript line 270 to 272, we have performed the further repeats and examined more TZPs of oocytes (n = 19-21) in GDF9 supplying experiments for extend the range of numbers (Fig. 5g-h). All of detected oocytes presented the same phenotype, which confirmed our findings that loss of Oo-Mv affected the formation of GC-TZPs.

Finally, the manuscript should be edited for clarity and accuracy. Again, a few examples:

Line 59: How is it known that the zona pellucida reduces the efficiency of OSFs?

Author response: Thanks for the valuable suggestion. We have changed the sentence to ‘but the existence of the ZP increases the physical distance between oocytes and GCs, which might reduce the efficiency of OSFs.’ in line 55 to 56.

Line 103: The observation that the head vesicles break down is interesting, but does not demonstrate a function.

Author response: Thanks for the valuable suggestion. We have changed the sentence to ‘showing complex cellular behaviors occurred in those Oo-Mv.’ in line 100 to 101.

Line 134: The statement that Ezrin and Moesin mRNA were ‘hardly detected’ in oocytes is not an accurate representation of the data shown in the Figure.

Author response: Thanks for the valuable suggestion. We have changed the representation to ‘Ezrin and Moesin, were not highly expressed in the oocytes’ in line 132.

Line 344: What is the direct evidence that the Oo-Mv regulate OSF release?

Author response: Thanks for the valuable suggestion. We have also realized that the evidence of Oo-Mv regulating OSF release was not convincing enough in the previous manuscript. Therefore, we have performed additional experiments and supplied further evidence to clarify the relationship between Oo-Mv and OSF release.

[Redacted]

[Redacted]

[Redacted]

Reviewer #3 (Remarks to the Author):

The authors have performed an extensive set of studies that have revealed a type of microvillus present on mouse oocytes that has a mushroom-shaped body at the distal end and that extends through most of the zona pellucida. They show that ablation of radixin in oocytes caused the loss of these microvilli. They also present evidence that the oocyte-secreted factor GDF9 was partly localized in these microvilli in the tip. Deleting radixin decreased the amount of oocyte-secreted factors and also decreased the numbers of transzonal processes from the granulosa cells. Females with radixin knocked down in their oocytes were subfertile and became infertile at an earlier age.

This is a very interesting and potentially very significant set of findings. The identification of a new anatomical structure in mammalian oocytes is particularly significant.

1) The main weakness is with the evidence connecting the effects of deleting oocyte radixin specifically with the loss of these types of microvilli. None of the experiments conclusively show that specifically ablating the long microvilli with the mushroom-shaped terminus is responsible for the reported effects on cumulus transzonal processes or female fertility.

Radixin is unlikely to be specific only to these long microvilli in oocytes. Indeed, in figure 2 e-h, it is clear that radixin protein is primarily localized in the cytoplasm (likely being trafficked to the membrane) and in a thick band at the cortex as well as in the long microvilli, with most in the cortical band. Phosphorylated radixin is localized primarily in the cortical band along with the long microvilli. When knocked out, all of this radixin disappears as expected. The identity of the cortical band is unclear. It is very well known that oocytes are densely covered with microvilli, which are presumably present in a greater number than the structures that are the focus of this report. Is the cortical band those microvilli? Or is in the subcortical actin band in oocytes? Perhaps both? The overexposure of this region to highlight the long microvilli makes it impossible to tell. Either way, the subcortical actin and/or the actin in the more numerous short microvilli could easily be disrupted by the loss of radixin and could account for the effects on cumulus cell transzonal processes and female fertility.

In summary, much stronger evidence is needed before it is claimed that these phenotypes are specifically due to the loss of the long microvilli. Other possibilities could be an effect on secretion, particularly of zona pellucida proteins or oocyte-derived factors at the oocyte surface, loss of small microvilli, a defect in zona pellucida biosynthesis, mislocalization of the egg's cortical granules, meiotic spindle mislocalization, or other effects.

Author response: Thanks for the valuable suggestion. We totally agree that the

concerns from reviewers about the relationship between Oo-Mv and OSF release, and the effects of *Radixin* deletion in oocytes in our study. Therefore, we have performed additional experiments and supplied further evidence to confirm these points.

[Redacted]

[Redacted]

To validate the effects of *Rdx* deletion in oocytes, we first detected the construction of oocyte cortex in *Oo-Rdx*^{-/-} females followed standard approach by labeling the actin with Alex488-phalloidin in oocytes^{1,2}. After staining, although less TZPs were found

in the ZP of oocytes from *Oo-Rdx*^{-/-} females compared to *Oo-Rdx*^{+/+} controls, an identical distribution and thickness of the oocyte cortex were observed in *Oo-Rdx*^{+/+} and *Oo-Rdx*^{-/-} oocytes (Fig. R3-2. a-b). This result showed that the deletion of *Rdx* had no effect on the formation of oocyte cortex.

Furthermore, we collected the superovulated oocytes from *Oo-Rdx*^{-/-} and *Oo-Rdx*^{+/+} females and performed the *in vitro* fertilization. Although less oocytes were collected from mutant females, all of these mutant oocytes were fertilized successful and formed the zygotes. Moreover, a comparable developmental capability of zygotes was found in *Oo-Rdx*^{+/+} and *Oo-Rdx*^{-/-} females till the blastocyst stage (Fig. R3-2. c-d). These results showed that the oocytes from *Oo-Rdx*^{-/-} females were fertile if they could pass the process of follicle selection. Therefore, we concluded that deletion of *Rdx* had no dramatic effects on the oocyte survival except the failure of Oo-Mv formation. With these data, we believed that deleting *Rdx* caused the Oo-Mv failure were the major if not the only phenotypes in the oocytes.

With these new data, we concluded that the oocytes control a timely and orderly release of OSFs to regulate the follicle development through forming unique Oo-Mv, and oocytic RDX mainly functions to construct the Oo-Mv to participate this progress.

Fig. R3-2. Deleting *Rdx* mainly affect the formation of Oo-Mv on oocytes. a-b, A normal formation of the oocyte cortex in *Oo-Rdx*^{-/-} female. Actin staining showing an abnormal formation

of GC-TZPs construction in Oo-*Rdx*^{-/-} oocytes, but an identical oocyte cortex was found in Oo-*Rdx*^{-/-} (n = 17) and Oo-*Rdx*^{+/+} (n = 18) oocytes. **c-d**, The *in vitro* fertilization of Oo-*Rdx*^{-/-} and Oo-*Rdx*^{+/+} oocytes. The superovulated oocytes of Oo-*Rdx*^{-/-} females and Oo-*Rdx*^{+/+} females were performed *in vitro* fertilization. No significantly difference in the rate of zygote formation and embryonic development was found. Data are presented as the mean ± SD with experiments performed in triplicate, and representative images are shown. Data were analyzed by 2-tailed unpaired Student's t-test and n.s. P ≥ 0.05. Scale bars in (a): 20 μm, in (c): 100 μm .

2) *The authors should specify the stage of oocyte shown in each figure. Some oocytes are at various stages of growth while some are fully-grown. The oocyte stage is only clearly indicated in fig 1d.*

Author response: Thanks for the valuable suggestion. Except Fig. 1d, all oocytes were at GV stage and were collected from late secondary or pre-antral follicles. We added more details in Line 451 to 452 of our M&M section in manuscript.

3) *The evidence showing that GDF9 is localized in the tips of the long microvilli is weak, depending on similar round structures being located in the same region of the zona pellucida and being of similar diameters. Colocalization with a marker for these long microvilli should be shown (e.g., radixin or mG).*

Author response: Thanks for the valuable suggestion. To further demonstrate the existence of GDF9 in the tips of Oo-Mv, we performed co-staining of RDX and GDF9 as reviewer's suggestion, and detected the expression of them through high-resolution imaging in the adult ovaries. As we showed in Fig. R3-3, the signal of GDF9 were clearly observed in the bubble of Oo-Mv in the ZP which were labeled by the RDX staining. This result clearly showed that GDF9 distributed in the tip of Oo-Mv, indicating OSFs were enriched in the vesicles of Oo-Mv to be released.

Fig. R3-3. GDF9 was localized in the vesicles of Oo-Mv. Immunostaining showing GDF9 protein (Red) localized in the oocytes and the vesicles of Oo-Mv (Green, labeled by RDX antibody). Scale bars: 5 μ m.

4) A more minor point is that the boundaries of the oocyte surface, the zona pellucida, and the innermost granulosa cells are not clear in the fluorescence micrographs. These should be indicated by labeling in the figures.

Author response: Thanks for the valuable suggestion. We labeled the boundaries of the oocyte surface or the innermost granulosa cells by white dot-line and ZP in Fig. 1b-c and e of manuscript.

Rebuttal References:

1. Chaigne A, Campillo C, Gov NS, Voituriez R, Azoury J, Umana-Diaz C, *et al.* A soft cortex is essential for asymmetric spindle positioning in mouse oocytes. *Nature cell biology* 2013, **15**(8): 958-966.
2. El-Hayek S, Yang Q, Abbassi L, FitzHarris G, Clarke HJ. Mammalian Oocytes Locally Remodel Follicular Architecture to Provide the Foundation for Germline-Soma Communication. *Curr. Biol.* 2018, **28**(7): 1124-1131 e1123.
3. Bouniol-Baly C, Hamraoui L, Guibert J, Beaujean N, Szollosi MS, Debey P. Differential transcriptional activity associated with chromatin configuration in fully grown mouse germinal vesicle oocytes. *Biol Reprod* 1999, **60**(3): 580-587.
4. Yan C, Wang P, DeMayo J, DeMayo FJ, Elvin JA, Carino C, *et al.* Synergistic roles of bone morphogenetic protein 15 and growth differentiation factor 9 in ovarian function. *Molecular endocrinology* 2001, **15**(6): 854-866.
5. Suzuki H, Ju J-C, Parks JE, Yang X. Surface Ultrastructural Characteristics of Bovine Oocytes Following Heat Shock. *Journal of Reproduction and Development* 1998, **44**(4): 345-351.
6. Reddy P, Liu L, Adhikari D, Jagarlamudi K, Rajareddy S, *et al.* Oocyte-specific deletion of Pten causes premature activation of the primordial follicle pool. *Science* 2008, **319**(5863): 611-613.
7. Chao, Yu, Yin-Li, Zhang, Wei-Wei, Pan, *et al.* CRL4 complex regulates mammalian oocyte survival and reprogramming by activation of TET proteins. *Science* 2013.
8. Hua, Zhang, Sanjiv, Risal, Nagaraju, Gorre, *et al.* Somatic Cells Initiate Primordial Follicle Activation and Govern the Development of Dormant Oocytes in Mice. *Current Biology* 2014.
9. Jiao X, Ke H, Qin Y, Chen ZJ. Molecular Genetics of Premature Ovarian Insufficiency. *Trends in endocrinology and metabolism* 2018, **29**(11): 795-807.
10. Dong JW, Albertini DF, Nishimori K, Kumar TR, Lu NF, Matzuk MM. Growth differentiation factor-9 is required during early ovarian folliculogenesis. *Nature* 1996, **383**(6600): 531-535.
11. Kikuchi S, Hata M, Fukumoto K, Yamane Y, Matsui T, Tamura A, *et al.* Radixin deficiency causes conjugated hyperbilirubinemia with loss of Mrp2 from bile canalicular membranes. *Nat Genet* 2002, **31**(3): 320-325.
12. Doi Y, Itoh M, Yonemura S, Ishihara S, Takano H, Noda T, *et al.* Normal development of mice and unimpaired cell adhesion cell motility actin-based cytoskeleton without compensatory up-regulation of ezrin or radixin in moesin gene knockout. *Journal Of Biological Chemistry* 1999, **274**(4): 2315-2321.
13. Chao CW, Chan DC, Kuo A, Leder P. The mouse formin (Fmn) gene: abundant circular RNA transcripts and gene-targeted deletion analysis. *Mol Med* 1998, **4**(9): 614-628.
14. Müller U. Ten years of gene targeting: targeted mouse mutants, from vector design to phenotype analysis. *Mech Dev* 1999, **82**(1-2): 3-21.
15. J., Peng, Q., Li, K., Wigglesworth, *et al.* Growth differentiation factor 9:bone morphogenetic protein 15 heterodimers are potent regulators of ovarian functions. *Proceedings of the National Academy of Sciences* 2013.
16. Higuchi CM, Maeda Y, Horiuchi T, Yamazaki Y. A Simplified Method for Three-Dimensional (3-D) Ovarian Tissue Culture Yielding Oocytes Competent to Produce Full-Term Offspring in Mice. *PLoS One* 2015, **10**(11): e0143114.
17. Green LJ, Shikanov A. In vitro culture methods of preantral follicles. *Theriogenology* 2016, **86**(1): 229-238.
18. Komatsu, K. & Masubuchi, S. Mouse oocytes connect with granulosa cells by fusing with cell membranes and form a large complex during follicle development. *Biol. Reprod.* **99**, 527-535 (2018).

Reviewers' Comments:

Reviewer #1:

Remarks to the Author:

The manuscript "Unique oocyte-derived microvilli control female fertility through optimising the ovarian follicle selection in mice" has improved in comparison to the first version. In this manuscript authors performed a series of experiments to demonstrate the role of Oo-Mv in the secretion of GDF9. According to the data RDX is responsible for generating mushroom like structures and the rupture of this gene causes GDF9 associated abnormalities. The manuscript is describing the role of Oo-Mv, which can impact follicle dynamics in different species including humans. The statistical analyses were well applied leading to interesting conclusions.

Based on the initial concerns raised in the first version authors were able to answer most of my concerns.

However, additional experiments are necessary:

- controls for GDF9 experiments.

In the Fig. R1-1. The GDF9 analysis in C demonstrate no differences. What type of negative control did you use? It is well know that oocyte present green autofluorescence. So my question is if the lack of difference could be influenced by the oocyte autofluorescence?

- better clarification of the phenotype. Is there co-localization of RDX and the Oo-Mv in the RDX-/- animals?

Minor concerns.

Line 393-394: "With our experimental data, we proposed that the role of Oo-Mv is an optimized system to select the qualified follicles, especially the early growing follicles." In order to affirm that should the author use a monovulatory species? How about the non-qualified follicles? Do they enter in atresia and do not display the Oo-Mv?

Reviewer #2:

Remarks to the Author:

Although the authors have added new data in the revised manuscript, the main concerns unfortunately remain. As stated in the first review, the data in Figs. 1 – 3 are very strong and clearly show that long oo-mv exist, that they are absent in oocytes lacking radixin, and that the targeted knockout of radixin in oocytes causes precocious loss of female fertility. The second part is where the problems lie. The authors argue that the role (or one role) of the oo-mv is to release GDF9. To support this, they attempt to show that GDF9 is present in the oo-mv and that the phenotype of the Rdx mutant is similar to the Gdf9 mutant. If correct, this introduces an entirely new concept into germ cell biology – oocytes have previously unknown long mv that bud off vesicles and these vesicles are the means by which an essential growth factor is delivered to the granulosa cells. Given the high stakes – a brand new concept – the authors need to provide substantial evidence to support their claim. Even in this revision, unfortunately, they have not done so.

First, do the oo-mv contain and release GDF9? The original manuscript showed high-resolution images of the oo-mv and foci of GDF9 (Extended Data Fig. 8 in the revision) in separate images. The request was to show co-localization. In the revised manuscript, the authors show a much less-clear image (Fig. 6a). There are two issues of concern here. One is the specificity of the antibody. The authors rely (Reporting summary) on the manufacturer's website. The image at that site shows staining of oocytes, but also of somatic cells that do not express GDF9. Even if the antibody

does recognize GDF9, the image in Fig. 6a has been obtained at a very high gain as indicated by the intense red staining of the oocyte. At this intensity, robust controls are needed to exclude that the signal seen in the oo-mv is not simply background. A control using an antibody against cytoplasmic protein not expected to be in the mv and imaged under the same conditions would be straightforward. Nor is quantification, such as frequency of co-localization, provided.

An additional significant concern is the use of injected labelled GDF9 protein (Fig. 6b) to track secretion. Secreted proteins, such as GDF9, are transported at the time of their translation into the lumen of the endoplasmic reticulum, where they are ultimately carried to the cell surface. It is difficult to see why or how an injected protein would be transported into the ER lumen. Unless the authors can show that injected GDF9 is transported into the ER lumen, the results shown in Fig. 6b do not tell us about the fate of GDF9 that is synthesized by the oocyte. In sum, the argument that the oo-mv release GDF9 rests entirely on the single image in Fig. 6a.

Other data provided to support a link between the oo-mv and GDF9 are confusing. The authors report that, in the Rdx mutants, there is more Gdf9 mRNA, less GDF9 protein in whole ovaries, and the same amount of GDF9 in oocytes (Fig. 6g, Fig. R1-1). No experimentally validated explanation is provided. They also report that injecting GDF9 into antrum of antral follicles reduces the number of oo-mv, whereas injecting anti-GDF9 increases the number (Fig. 6e). The aim of this experiment was not clear to the reviewer. The authors conclude (l. 323) that the formation of oo-mv is directly related to oocyte-secreted factors, like GDF, which is entirely separate from the idea that the oo-mv regulate GDF9 release.

Second, does the phenotype resemble the Gdf9 knockout phenotype? The authors argue that the Rdx mutant may not fully suppress GDF9 release, thus accounting for the phenotypic differences. This is possible, but the concern is that the only substantive similarity between the two is that both ultimately lead to infertility.

- i) Multilayer follicles develop in the Rdx mutant (Fig. 3b, 3d, 5c), but not in the Gdf9 mutant (published work; also Fig. R2-4)
- ii) The Rdx mutant is initially fertile (Fig. 3e), whereas the Gdf9 mutant is not.
- iii) The decrease in the Rdx mutant of abundance of GDF9 target genes (Extended Data Fig. 7) is modest and limited to only two of the four targets tested.
- iv) Oocytes in the Rdx mutant grow more slowly than wild-type in vitro (Fig. 4d), whereas they reach a larger final size than wild-type in Gdf9 mutants.
- v) The authors state that Matzuk group reported that Bmp15 $-/-$; Gdf9 $+/-$ females showed "decreased fertility in young adult and failed to produce any offspring after 6-month of mating, which is consistent with the ... Rdx mutants." The Matzuk group reported that on one genetic background, the double-mutants produced fewer and smaller litters; on a different background, they produced no litters even after being caged with males for 6 months. They did not report an age-linked reduction in litter frequency or size for any condition. Thus, one cannot conclude that the phenotype matches the Rdx mutant.

Additional concerns:

In response to the concern that the cortex of the Rdx $-/-$ oocytes may be abnormal (and so could be the reason for the reduced fertility), the authors show that the thickness of the actin band is the same in $+/+$ and $-/-$ oocytes (Extended Data Fig. 4). This does not show that the cortex is normal - just that one element, assessed rather superficially, seems to be so. The authors also argue that since the $-/-$ oocytes can give rise to blastocysts, the radixin in the cortex is not required for normal oocyte development. But this observation argues equally strongly that the oo-mv are not necessary.

Lack of methodological detail continues to be a concern. Two examples: First, the authors describe culturing ovarian fragments from 21-day mice. These likely contained antral follicles, though it is not specified. Responding to the request for more information, the authors have added that the

fragments were cultured in Matrigel as described in three publications. Only one of the three actually details the conditions that were used, and follicles at an earlier stage of development were cultured. Second, they state that freshly collected oocytes were stained using phalloidin. Phalloidin cannot pass through intact cell membranes – the cells must be fixed and permeabilized. Although these may seem like minor points, the concern is the information supplied in the manuscript is not enough to enable the experiments to be reproduced.

Reviewer #3:

Remarks to the Author:

The authors have shown colocalization of radixin and GDF9 in the termini of the long microvilli and have shown that GDF9 is released in the same region where the tips of these microvilli lie, providing better evidence that GDF9 is secreted from these structures. They have also shown that GDF9 regulates the formation of granulosa-derived transzonal processes. They have also addressed the more minor points raised.

Rebuttal letter:

Reviewer #1 (Remarks to the Author):

The manuscript “Unique oocyte-derived microvilli control female fertility through optimising the ovarian follicle selection in mice” has improved in comparison to the first version. In this manuscript authors performed a series of experiments to demonstrate the role of Oo-Mv in the secretion of GDF9. According to the data RDX is responsible for generating mushroom like structures and the rupture of this gene causes GDF9 associated abnormalities. The manuscript is describing the role of Oo-Mv, which can impact follicle dynamics in different species including humans. The statistical analyses were well applied leading to interesting conclusions.

Based on the initial concerns raised in the first version authors were able to answer most of my concerns.

However, additional experiments are necessary:

- controls for GDF9 experiments.

In the Fig. R1-1. The GDF9 analysis in C demonstrate no differences. What type of negative control did you use? It is well know that oocyte present green autofluorescence. So my question is if the lack of difference could be influenced by the oocyte autofluorescence?

Author response: Thank you so much for your valuable suggestions. To clarify the localization of GDF9, we had tested the immunostaining of GDF9 with several different anti-GDF9 antibodies, including the antibodies from R&D, AF739 (Fig. R1-1A, top panel), Santa Cruz, sc-514933 (R1-1A, middle panel) and Abcam, ab38544 (R1-1A, bottom panel) with the basic negative control of related IgG for each antibody (Fig. R1-1A, right panel) in our previous experiments. To further verify the specificity of these GDF9 antibodies from different vendors, ovarian sections from *Gdf9* knockout (*Gdf9*^{-/-}) mice were stained simultaneously with the wild-type ovarian sections using the same protocols (Fig. R1-1B). As a result, we found that the antibody from R&D (cat#: AF739; Lot#: DRY0418101, 20 µg/ml, 1:50) was suitable for detecting the expressions of GDF9 in oocytes, which gave a clearly positive staining of GDF9 in the wild-type oocytes but detected no signal in the oocyte of *Gdf9*^{-/-} ovaries (Fig. R1-1, top panel). Therefore, we used this antibody to detect the GDF9 expressions in the Fig. 6a. We are sorry we didn't supply these control data in the previous version of rebuttal letter and manuscript.

Fig. R1-1. Testing the specificity of different anti-GDF9 antibodies by immunostaining GDF9 in wild-type and *Gdf9*^{-/-} ovarian sections. (A) GDF9 expression was detected by different GDF9 antibodies in the wild-type ovarian sections (PD35), and the related IgGs were used as the negative control. Only the R&D antibody showed a strong positive signal in the oocytes and weak reaction in the GCs. (B) To test the specificity of antibodies, ovarian sections from *Gdf9*^{-/-} females at PD35 were stained with different GDF9 antibodies. No positive signal was detected in the *Gdf9*^{-/-} oocytes by the R&D antibody (top panel), thus proving it to be the specific antibody suitable for IF staining of GDF9 expression in mice. Scale bars: 20 μ m.

- better clarification of the phenotype. Is there co-localization of RDX and the Oo-Mv in the RDX^{-/-} animals?

Author response: Thank you so much for your valuable suggestions. As we showed in Fig. 3b (*Rdx* KO efficiency testing) in the manuscript, there was no expressions of both RDX and p-RDX in the Oo-*Rdx*^{-/-} ovaries. To clarify the phenotype, we re-checked the staining results carefully and found that neither the expression of RDX nor the formation of Oo-Mv was observed in the oocytes of Oo-*Rdx*^{-/-} females.

Minor concerns.

Line 393-394: “With our experimental data, we proposed that the role of Oo-Mv is an optimized system to select the qualified follicles, especially the early growing

follicles.” In order to affirm that should the author use a monovulatory species? How about the non-qualified follicles? Do they enter in atresia and do not display the Oo-Mv?

Author response: Dear reviewer, thank you so much for your kindly help and valuable suggestions. This is a very interesting and important point about the potential role of Oo-Mv in monovulatory species including the primate. Based on our experimental evidence, we believed that the major function of Oo-Mv was to participate the process of follicle selection, especially the selection of early growing follicles before the antral stage. Therefore, we guessed that the non-qualified follicles with less Oo-Mv should go to atresia, and the qualified follicles would develop to further stage for the final selection during cyclic recruitment to form the dominant follicle in monovulatory female.

Reviewer #2 (Remarks to the Author):

Although the authors have added new data in the revised manuscript, the main concerns unfortunately remain. As stated in the first review, the data in Figs. 1 – 3 are very strong and clearly show that long oo-mv exist, that they are absent in oocytes lacking radixin, and that the targeted knockout of radixin in oocytes causes precocious loss of female fertility. The second part is where the problems lie. The authors argue that the role (or one role) of the oo-mv is to release GDF9. To support this, they attempt to show that GDF9 is present in the oo-mv and that the phenotype of the Rdx mutant is similar to the Gdf9 mutant. If correct, this introduces an entirely new concept into germ cell biology – oocytes have previously unknown long mv that bud off vesicles and these vesicles are the means by which an essential growth factor is delivered to the granulosa cells. Given the high stakes – a brand new concept – the authors need to provide substantial evidence to support their claim. Even in this revision, unfortunately, they have not done so.

Author response: Dear reviewer, we're really grateful for your valuable comments. Your appreciation of the scientific significance of our finding is extremely encouraging, and is really honorable to us - a young group. As we responded in the previous revision, we totally agree to your concern about the relationship between GDF9 (OSFs) and the functional role of Oo-Mv. We therefore performed a series of additional experiments to clarify this point. In this revision, we have supplied robust controls for the GDF9 localization and other new data to support this point. Hoping these data will erase the concerns.

First, do the oo-mv contain and release GDF9? The original manuscript showed high-resolution images of the oo-mv and foci of GDF9 (Extended Data Fig. 8 in the revision) in separate images. The request was to show co-localization. In the revised manuscript, the authors show a much less-clear image (Fig. 6a). There are two issues of concern here. One is the specificity of the antibody. The authors rely (Reporting summary) on the manufacturer's website. The image at that site shows staining of oocytes, but also of somatic cells that do not express GDF9. Even if the antibody does recognize GDF9, the image in Fig. 6a has been obtained at a very high gain as indicated by the intense red staining of the oocyte. At this intensity, robust controls are needed to exclude that the signal seen in the oo-mv is not simply background. A control using an antibody against cytoplasmic protein not expected to be in the mv and imaged under the same conditions would be straightforward. Nor is quantification, such as frequency of co-localization, provided.

Author response: Apparently, we misunderstood your previous suggestion on the co-localization of Oo-Mv and GDF9 expression. We apologize sincerely. Our original experimental design of R-GDF9 tracing was to inject the rhodamin-labeled GDF9 (R-GDF9) into the *Gdf9-Cre;mTmG* oocytes in which the Oo-Mv was labeled in green by mG fluorescence protein. However, the granulosa cell-derived TZPs (GC-TZPs)

labeled in red by mTomato (mT) were kept in the zona pellucida (ZP) (Fig. R2-1). Because the mTomato (mT) and rhodamin share the same laser emission (555 nm) spectrum, the granulosa cell-derived TZPs (GC-TZPs) labeled in red by mTomato (mT) were also able to be captured simultaneously when imaging the R-GDF9 was conducted. The R-GDF9 would then be masked by the stronger mTomato (mT) signal existed in the ZP of oocytes. Therefore, we traced R-GDF9 in WT oocytes and compared the R-GDF9 spots in WT oocytes with the size of Oo-Mv vesicles in *Gdf9-Cre;mTmG* oocytes (see Extended Data Fig. 9.).

Fig. R2-1. The GC-TZPs with strong mT signal were captured in the ZP of *Gdf9-Cre;mTmG* oocyte.

The Oo-Mv labeled by mG were captured in green, whereas mT channel (555 nm) labeled GC-TZPs detected in the ZP of *Gdf9-Cre;mTmG* oocyte were captured in red. Scale bars: 5 μ m.

To clarify the localization of GDF9, we had tested the immunostaining of GDF9 with several different anti-GDF9 antibodies, including the antibodies from R&D, AF739 (Fig. R2-2A, top panel), Santa Cruz, sc-514933 (R2-2A, middle panel) and Abcam, ab38544 (R2-2A, bottom panel) with the basic negative control of related IgG for each antibody (Fig. R2-2A, right panel) in our previous experiments. To further verify the specificity of these GDF9 antibodies from different vendors, ovarian sections from *Gdf9* knockout (*Gdf9*^{-/-}) mice were stained simultaneously with the wild-type ovarian sections using the same protocols (Fig. R2-2B). As a result, we found that the antibody from R&D (cat#: AF739; Lot#: DRY0418101, 20 μ g/ml, 1:50) was suitable for detecting the expressions of GDF9 in oocytes, which gave a clearly positive staining of GDF9 in the wild-type oocytes but detected no signal in the oocyte of *Gdf9*^{-/-} ovaries (Fig. R2-2, top panel). Therefore, we used this antibody to detect the GDF9 expressions in the Fig. 6a. We are sorry we didn't supply these control data in the previous version of rebuttal letter and manuscript.

Fig. R2-2. Testing the specificity of different anti-GDF9 antibodies by immunostaining GDF9 in wild-type and *Gdf9*^{-/-} ovarian sections. (A) GDF9 expression was detected by different GDF9 antibodies in the wild-type ovarian sections (PD35), and the related IgGs were used as the negative control. Only the R&D antibody showed a strong positive signal in the oocytes and weak reaction in the GCs. (B) To test the specificity of antibodies, ovarian sections from *Gdf9*^{-/-} females at PD35 were stained with different GDF9 antibodies. No positive signal was detected in the *Gdf9*^{-/-} oocytes by the R&D antibody (top panel), thus proving it to be the specific antibody suitable for IF staining of GDF9 expression in mice. Scale bars: 20 μ m.

Even if the antibody does recognize GDF9, the image in Fig. 6a has been obtained at a very high gain as indicated by the intense red staining of the oocyte. At this intensity, robust controls are needed to exclude that the signal seen in the oo-mv is not simply background. A control using an antibody against cytoplasmic protein not expected to be in the mv and imaged under the same conditions would be straightforward. Nor is quantification, such as frequency of co-localization, provided.

Author response: Meanwhile, we also added new controls of high-resolution GDF9 staining to confirm the specificity of GDF9 localization in Fig. 6a, Extended Data Fig. 8 and Fig. R2-3A. Beside the IgG control (Fig. R2-3B), the ovarian sections of *Gdf9*^{-/-} females were stained with GDF9 antibody (R&D, 1:50) under the same procedure to serve as the "genuine" negative control. Indeed, in the *Gdf9*^{-/-} oocytes, there was no specific signal of GDF9 staining (red) found in the Oo-Mv vesicles that were stained in green by the RDX antibody (Fig. R2-3B, arrows). This result therefore confirmed the

specificity of high-resolution imaging of GDF9 localization in the wild-type ovaries (Fig. R2-3A), and the data of controls was also added into the manuscript in Extended Data Fig. 8.

As reviewer's suggestion, we next quantified the ratio of co-localization of GDF9 and RDX labeled vesicles in oocytes. As a result, the Oo-Mv with GDF9 localization was observed in all detected oocytes (n = 29) in PD35 wild-type ovaries, and the statistical analysis showed that around 70% (71.5 ± 4.2 %) of Oo-Mv were GDF9 positive (Fig. R2-3C). In sharp contrast, none Oo-Mv had the positive signal of GDF9 in IgG control and *Gdf9*^{-/-} oocytes (Fig. R2-3C).

Followed the suggestion from reviewer, we chose several well-identified oocyte cytoplasm expressing proteins, including DDX4 (Dead-box helicase 4, ab27591, Abcam)¹, DAZL (Deleted in Azoospermia-like, GTX89448, GeneTex)², MATER (maternal antigen that embryos require, also known as NLRP5, generated by Abmart, which is a kindly gift from Dr. Lei Li)³ to perform the immunostaining with high-resolution detection. Interestingly, we found two different expressing models of these proteins. As predicted by reviewer, the localization of MATER was clearly observed in the cytoplasm of oocytes, but was not detected in the Oo-Mv, showing a selective localization of oocyte proteins in the vesicles of Oo-Mv. In contrast, the expressions of DDX4 and DAZL were observed in both the oocyte cytoplasm and the vesicles of Oo-Mv. These data suggested that the vesicles of Oo-Mv contained the oocyte cytoplasm, and our high-resolution imaging approach was sensitive enough to identify the specific expression of oocyte proteins in the vesicles (Fig. R2-3D).

Fig. R2-3. High-resolution detection of the GDF9 in the vesicles of Oo-Mv.

(A) Immunostaining showing GDF9 protein (Red) localized in the oocytes and the vesicles (arrowheads) of Oo-Mv (Green, labeled by RDX antibody). (B) Different controls showing the specificity of GDF9 staining. The negative controls including the IgG and *Gdf9*^{-/-} ovarian sections showed that no specific signal in the vesicles of Oo-Mv after staining (arrows). (C) The quantitation of the ratio with GDF9 signal in Oo-Mv vesicles of different wild-type oocytes (n = 29), *Gdf9*^{-/-} oocytes (n = 10) and IgG signal in oocytes (n = 8). (D) As the controls of oocyte cytoplasm localized protein, the expression of MATER was only observed in the cytoplasm of oocytes, but DDX4 and DAZL were detected in both the oocyte cytoplasm and the vesicles (arrowheads) of Oo-Mv. No signal in the related IgG controls showing the specificity of the immunostaining (arrows). Scale bars: 5 μ m.

An additional significant concern is the use of injected labelled GDF9 protein (Fig. 6b) to track secretion. Secreted proteins, such as GDF9, are transported at the time of their translation into the lumen of the endoplasmic reticulum, where they are ultimately carried to the cell surface. It is difficult to see why or how an injected protein would be transported into the ER lumen. Unless the authors can show that injected GDF9 is transported into the ER lumen, the results shown in Fig. 6b do not

tell us about the fate of GDF9 that is synthesized by the oocyte. In sum, the argument that the oo-mv release GDF9 rests entirely on the single image in Fig. 6a.

Author response: Thanks for your comments. We agreed that the injection of exogenous GDF9 was not a perfect approach to trace the fate of oocyte synthesized OSFs, but this experiment still supplied directly visualized evidence showing that the Oo-Mv was functional to mediate the release of GDF9. Indeed, the secretory proteins are transported through the endoplasmic reticulum to the plasma membrane. In our tracing experiments, we observed a specific aggregative distribution of fluorescent signal in R-GDF9 group, which was in sharp contrast to a diffuse and uniform fluorescent distribution in the oocytes after the R-BSA or rhodamine-only injection (Fig. 6b in the manuscript and Fig. R2-4). This pattern of localization was similar as the previous reported ER localization in oocytes in publications^{4, 5}. We guessed that the injected R-GDF9 might be bound to ER for transporting with unknown mechanisms.

Fig. R2-4. The distribution of R-GDF9 was similar with the ER localization in oocytes in previous publications. (A) As the negative control, R-BSA showed a diffuse and uniform fluorescent distribution in oocytes. (B) After microinjection, R-GDF9 showed an aggregative distribution and formed the cloudy structure in oocytes. (C) Distribution of ER in oocytes stained by ER-tracker or DiI in the previous publications^{4, 5}. Scale bars in (A-B): 25 μ m.

With reviewer's suggestion and the observation of aggregative R-GDF9 distribution in our results, we next detected the localization of ER in GV oocytes by ER-tracker (Beyotime, C1041) with the high-resolution imaging approach. As expected, the ER

exhibited an aggregative and cloudy distribution (Fig. R2-5A, middle) in the oocyte cytoplasm, which was consistent with the previously reports^{4,5} (Fig. R2-4C and 2-5A). Interestingly, we found that there were lots of ER tracker labeled bubbles (ER-bubbles) in the ZP (Fig. R2-5B, middle, arrowheads) in our high-resolution imaging, which was similar as the distributions of Oo-Mv (Fig. R2-5B, bottom, arrowheads) and the R-GDF9 spots (Fig. R2-5B, top, arrowheads) in the study. Meanwhile, the counting results showed a comparable number of ER-bubbles (107 ± 6 , $n = 17$) and mG labeled vesicles (110 ± 7 , $n = 14$) on Oo-Mv of *Gdf9-Cre;mTmG* ovaries (Fig. R2-5C). Moreover, the quantitation of the fluorescent intensity showed that these ER-bubbles exhibited a significant higher signal intensity than that in the oocyte cytoplasm (Fig. R2-5D and E). These results suggested that an enrichment of ER-transported protein in the vesicles of Oo-Mv, which was in consistent to our conclusion about the OSF release function of Oo-Mv. With these new data, we proposed that the OSFs were translated into the endoplasmic reticulum in oocyte cytoplasm, where they were ultimately carried to the cell surface especially the Oo-Mv to be released to the intercellular space between oocyte and GCs. The related data was also added in the manuscript in line 315-326 and Fig. 6e-g and Extended Data Fig. 10.

Fig. R2-5. A similar distribution of ER with R-GDF9 and Oo-Mv in the living oocytes.

(A) ER-tracker staining showed a cloudy ER distribution (middle) in the oocyte cytoplasm, and many ER bubbles surrounding oocytes which were similar with the R-GDF9 distribution (top) and Oo-Mv vesicles (bottom). (B) The magnified images showing the ER-bubbles (middle, arrowheads) surrounding the oocyte, which was comparable to R-GDF9 spots (top, arrowheads) and Oo-Mv vesicles (bottom, arrowheads). (C) Quantification of the number of R-GDF9 spots ($n = 11$), ER-bubbles ($n = 17$) and Oo-Mv vesicles ($n = 14$). (D) Line scan of fluorescent intensities by Image J software showing a high fluorescent intensity of ER signal in bubbles compared to that in oocyte

cytoplasm. The Y-axis was corresponded to the red line in cutting ER staining image. (E) Quantification of the fluorescence intensity of ER signal in oocyte cytoplasm and bubbles (n = 20). Data are presented as the mean \pm SD with experiments performed in triplicate, and representative images are shown. Data were analyzed by 2-tailed unpaired Student's t-test and *** P < 0.001, n.s. P \geq 0.05. Scale bars in (A) and (B): 20 μ m; in (E): 10 μ m .

Other data provided to support a link between the oo-mv and GDF9 are confusing. The authors report that, in the Rdx mutants, there is more Gdf9 mRNA, less GDF9 protein in whole ovaries, and the same amount of GDF9 in oocytes (Fig. 6g, Fig. R1-1). No experimentally validated explanation is provided. They also report that injecting GDF9 into antrum of antral follicles reduces the number of oo-mv, whereas injecting anti-GDF9 increases the number (Fig. 6e). The aim of this experiment was not clear to the reviewer. The authors conclude (l. 323) that the formation of oo-mv is directly related to oocyte-secreted factors, like GDF, which is entirely separate from the idea that the oo-mv regulate GDF9 release.

Author response: As we described in the previous revision, we totally agree to your concern about the relationship between GDF9 (OSFs) and the functional role of Oo-Mv. We therefore performed the experiments including the identification of *Gdf9*/GDF9 expressions and the disruptions of GDF9 function in follicles in the revised manuscript. In the Oo-Mv counting experiments of both *Gdf9* KO follicles and the anti-GDF9 antibody injected follicles, a comparable phenotype that the increase of Oo-Mv number was observed with the disruption of GDF9 function. In contrast, the number of Oo-Mv was significantly decreased when extra GDF9 was injected into the cavity of antral follicles. These data showed a negative correlation between the concentration of OSF in the intercellular space and the number of Oo-Mv on oocytes, which suggested a feedback loop of the Oo-Mv formation and OSF release. We believed that these data also supplied important evidence to support the function of Oo-Mv in regulating the OSF release.

Although these data were supplied the evidence to connect the relationship between Oo-Mv and OSFs, we agreed the reviewer's suggestion that these experiments were not directly related to the major conclusion in the manuscript. We therefore have removed related data from the current version of revised manuscript (Fig. 6e-f in the previous revised manuscript).

Second, does the phenotype resemble the Gdf9 knockout phenotype? The authors argue that the Rdx mutant may not fully suppress GDF9 release, thus accounting for the phenotypic differences. This is possible, but the concern is that the only substantive similarity between the two is that both ultimately lead to infertility.

i) Multilayer follicles develop in the Rdx mutant (Fig. 3b, 3d, 5c), but not in the Gdf9 mutant (published work; also Fig. R2-4)

ii) The Rdx mutant is initially fertile (Fig. 3e), whereas the Gdf9 mutant is not.

iii) *The decrease in the Rdx mutant of abundance of GDF9 target genes (Extended Data Fig. 7) is modest and limited to only two of the four targets tested.*

iv) *Oocytes in the Rdx mutant grow more slowly than wild-type in vitro (Fig. 4d), whereas they reach a larger final size than wild-type in Gdf9 mutants.*

v) *The authors state that Matzuk group reported that Bmp15 -/-; Gdf9 +/- females showed “decreased fertility in young adult and failed to produce any offspring after 6-month of mating, which is consistent with the ... Rdx mutants.” The Matzuk group reported that on one genetic background, the double-mutants produced fewer and smaller litters; on a different background, they produced no litters even after being caged with males for 6 months. They did not report an age-linked reduction in litter frequency or size for any condition. Thus, one cannot conclude that the phenotype matches the Rdx mutant.*

Author response: Thanks for your suggestions. As we stated from the first version of the manuscript, the Oo-*Rdx*^{-/-} represented a series of phenotypes including the lack of Oo-Mv on oocytes, the accelerated loss of follicles and a subfertile in females. Therefore, these phenotypes didn't resemble the infertile phenotype of *Gdf9*^{-/-} females.

Based on our experimental finding, we concluded that Oo-Mv played as a regulating role rather than an indispensable structure to control the OSF release, as we presented in the manuscript. First of all, as a newly identified structure, the Oo-Mv exhibits a unique structure in the ZP. They do not attach to the GC's membrane, but forms a swollen vesicle tip which is close to the GCs. These structural features indicated the Oo-Mv play as a secreting structure rather than a direct connection to participate the communications. Secondary, the results of the existence of OSFs (GDF9), the high intensity of ER-transported protein in the bubbles of ZP, and the exogenous OSFs released through Oo-Mv have clarified our hypothesis that the functional role of Oo-Mv was to regulate the release of OSFs on oocytes. Finally, the phenotypes of Oo-*Rdx*^{-/-} represented that the lack of Oo-Mv led to an accelerated follicle loss, a retardation of oocyte and follicle growth, confirmed that loss of Oo-Mv affected the follicle development. However, the subfertile phenotype also showed that the follicle development was not completely suppression after loss of Oo-Mv, and the successful IVF of superovulated oocytes from Oo-*Rdx*^{-/-} females confirmed that loss of Oo-Mv had no effect on the oocyte survival if they developed to antral stage. Therefore, we concluded that the major function of Oo-Mv is to make the release of OSFs timely and orderly, guaranteeing a precise and efficient regulation to determine the developmental fate of ovarian follicles.

With the valuable suggestions from reviewers, we believed we had supplied systematic and logical experimental data to support the conclusions. It seems that the oocyte/Oo-Mv—OSFs—GC/TZPs construct a completely regulating loop in controlling the development of early follicles, and the OSFs especially the GDF9 plays a central role in this regulating loop.

For the phenotype of *Bmp15*^{-/-}; *Gdf9*^{+/-} females, we are sorry that we only focused the phenotype in one background. As reviewer's suggestion, we didn't add the related discussion in the manuscript.

Additional concerns:

In response to the concern that the cortex of the Rdx -/- oocytes may be abnormal (and so could be the reason for the reduced fertility), the authors show that the thickness of the actin band is the same in +/+ and -/- oocytes (Extended Data Fig. 4). This does not show that the cortex is normal - just that one element, assessed rather superficially, seems to be so. The authors also argue that since the -/- oocytes can give rise to blastocysts, the radixin in the cortex is not required for normal oocyte development. But this observation argues equally strongly that the oo-mv are not necessary.

Author response: Thanks for your suggestions. To confirm the phenotype of normal oocyte cortex in Oo-*Rdx*^{-/-} females, we discussed with several experts in the field of oocyte maturation, including Dr. Lei Li in CAS, Dr. Bo Xiong in Nanjing Agricultural University and Dr. You-qiang Su in Nanjing Medical University. With their suggestions, we performed a LCA-FITC (FITC-labelled lens culinaris agglutinin, ThermoFisher Scientific, L32475) staining to identify the distributions of cortical granules in oocytes^{6, 7}. As a result, we found that a normal distribution and fluorescent intensity of cortical granules in Oo-*Rdx*^{-/-} oocytes compared to control (Fig. R2-6). This result supported our conclusion that deletion of *Rdx* had no significant effect on the formation of oocyte cortex.

For the function of Oo-Mv in oocyte development, our experimental evidence showed that the failure of Oo-Mv formation in Oo-*Rdx*^{-/-} oocytes had no dramatic effects on the oocyte maturation and fertilization. However, loss of Oo-Mv led to a series of defects on follicle development in Oo-*Rdx*^{-/-} females, especially the early development of follicles. In details, an accelerated loss of follicles was observed in the Oo-*Rdx*^{-/-} ovaries, and the peak of follicle loss occurred before the adulthood when hormone independent follicle growth was active. Meanwhile, the cell proliferating and apoptotic analysis showed that the major defects of follicle growth were observed in the primary and secondary stages, which were the major OSF-dependent stages of folliculogenesis. These defects finally led to a subfertile phenotype in females. With these experimental facts, we concluded that the major function of Oo-Mv is to make the release of OSFs timely and orderly rather than to guarantee the survival of oocytes. We are sorry that we didn't state our point clearly in the previous rebuttal letter.

Fig. R2-6. Detecting the cortical granules in oocytes by LCA-FITC in *Oo-Rdx*^{-/-} and control oocytes.

(A) LCA-FITC immunostaining showed a normal cortical granules distribution in *Oo-Rdx*^{-/-} compared to control oocytes. (B) The quantitation analysis showed no significantly difference of the cortical granule fluorescent intensity in *Oo-Rdx*^{-/-} (n = 40) and control oocytes (n = 21). Data are presented as the mean ± SD with experiments performed in triplicate, and representative images are shown. Data were analyzed by 2-tailed unpaired Student's *t*-test and n.s. P ≥ 0.05. Scale bars: 50 μm.

Lack of methodological detail continues to be a concern. Two examples: First, the authors describe culturing ovarian fragments from 21-day mice. These likely contained antral follicles, though it is not specified. Responding to the request for more information, the authors have added that the fragments were cultured in Matrigel as described in three publications. Only one of the three actually details the conditions that were used, and follicles at an earlier stage of development were cultured. Second, they state that freshly collected oocytes were stained using phalloidin. Phalloidin cannot pass through intact cell membranes – the cells must be fixed and permeabilized. Although these may seem like minor points, the concern is the information supplied in the manuscript is not enough to enable the experiments to be reproduced.

Author response: Thank you so much for the suggestions to improve our work. For the phalloidin staining, we followed previous protocol strictly⁸ and the live oocytes were fixed with 4% PFA firstly in our study, we had added this detail in the method. Sorry for the confusing of the methodology, and we have expanded all methods in details especially the method of the follicle culture. Meanwhile, the detail protocols of the detection of Oo-Mv in *Gdf9-Cre;mTmG* females and the ovarian follicle culture were expanded followed the format of the Protocol Exchange, which was also attached at the end of the rebuttal letter. Thanks again for your kindly and patient suggestions to improve our study.

Reviewer #3 (Remarks to the Author):

The authors have shown colocalization of radixin and GDF9 in the termini of the long microvilli and have shown that GDF9 is released in the same region where the tips of these microvilli lie, providing better evidence that GDF9 is secreted from these structures. They have also shown that GDF9 regulates the formation of granulosa-derived transzonal processes. They have also addressed the more minor points raised.

Author response: Thanks for the kindly help. We really appreciated your valuable suggestions to help us improving our study.

Rebuttal Method

High resolution GDF9, DDX4, DAZL and MATER staining

The wild-type or *Gdf9*^{-/-} PD35 ovarian sections were deparaffinized, rehydrated, and subjected to high-temperature (95-98°C) antigen retrieval for 16 mins with 0.01% sodium citrate buffer (pH 6.0). The sections were then blocked with 10% normal donkey serum (Jackson ImmunoResearch, 017-000-121) for 60 mins at room temperature and incubated with primary antibodies overnight at 4°C. Subsequently, the sections were incubated with fluorophore-conjugated donkey secondary antibodies (1:200, Life Technologies) for 2 h at 37°C. The homologous IgGs with the primary antibodies were used to be the negative control to guarantee the specificity of staining.

To detect GDF9 protein expression in the Oo-Mv, we incubated sections with GDF9 antibody (goat, 1:50, AF739, R&D) and the RDX antibody (rabbit, 1:100, ab52495, Abcam) for labeled Oo-Mv. To detect oocyte cytoplasm protein DDX4, DAZL and MATER expression in the Oo-Mv, we incubated sections with DDX4 (mouse, 1:100, ab27591, Abcam) or DAZL (goat, 1:100, GTX89448, GeneTex) or MATER (mouse, 1:100, generated by Abmart, which is a kindly gift from Dr. Lei Li in CAS) and the RDX antibody (rabbit, 1:100, ab52495, Abcam) for labeling Oo-Mv.

For the high-resolution detections of GDF9, DDX4, DAZL or MATER, the co-localization of GDF9, DDX4, DAZL or MATER with vesicles of Oo-Mv were acquired as a z-stack model with an optical slice thickness of 0.3 μm after co-staining of GDF9, DDX4, DAZL or MATER with RDX, and confocal sections covered ~8 μm tissue thickness. Both the IgGs and the ovarian sections from *Gdf9* KO females were used as the negative control to guarantee the specificity of staining. The images were acquired by an Andor Dragonfly spinning-disc confocal microscope equipped with a 100×, 1.44 N.A. oil objective (Leica HC PL APO), a scientific complementary metal-oxide semiconductor (sCMOS) camera (Andor Zyla 4.2), and the 488-nm (anti-RDX, Oo-Mv) and 568-nm (anti-GDF9, DDX4, DAZL and IgGs) lines of the Andor Integrated Laser Engine (ILE) system with a spinning-disc confocal scan head (Andor Dragonfly 500). The images were acquired through z-step mode with the index. In detail, images were acquired with laser 488-nm and laser 568-nm around 10-15%, exposure time 100-200 ms. Images were acquired by Fusion 2.1 software (<https://andor.oxinst.com/products/dragonfly#fusion>).

ER-tracker staining and analysis

To observe the ER distribution of oocytes, denuded living oocytes at GV stage (average diameter around 65 μm) were incubated with ER-tracker (1:1000, Beyotime, C1041) for 30 mins at 37°C, 5% CO₂. After washing with PBS, the stained oocytes were imaged by an Andor Dragonfly spinning-disc confocal microscope as previous described indexes. For the detection of ER fluorescent intensity of oocyte cytoplasm and bubbles, the single Z of mean ER fluorescent intensity of oocyte cytoplasm and bubbles were measured by Image J software.

LCA-FITC staining

To observe the cortical granules of oocytes, Oo-*Rdx*^{+/+} and Oo-*Rdx*^{-/-} oocytes at GV stage (average diameter around 65 μm) were fixed in 4% PFA in PBS for 15 mins and permeabilized in 0.5% Triton-X-100 for 20 mins at room temperature. Then, oocytes were blocked with 1% BSA (Sigma-Aldrich, V900933) in PBS for 1 h and incubated with LCA-FITC (1:100 dissolved in PBS, ThermoFisher Scientific, L32475, a gift from Dr. Bo Xiong in Nanjing Agricultural University, China) 2 h at room temperature⁶. After washing with PBS, the stained oocytes were imaged by an Andor Dragonfly spinning-disc confocal microscope as previous described indexes.

Reference

1. Toyooka Y, Tsunekawa N, Takahashi Y, Matsui Y, Satoh M, Noce T. Expression and intracellular localization of mouse Vasa-homologue protein during germ cell development. *Mech. Dev.* 2000, **93**(1-2): 139-149.
2. Ruggiu M, Speed R, Taggart M, McKay SJ, Kilanowski F, Saunders P, *et al.* The mouse Dazl gene encodes a cytoplasmic protein essential for gametogenesis. *Nature* 1997, **389**(6646): 73-77.
3. Yu XJ, Yi Z, Gao Z, Qin D, Zhai Y, Chen X, *et al.* The subcortical maternal complex controls symmetric division of mouse zygotes by regulating F-actin dynamics. *Nat. Commun.* 2014, **5**: 4887.
4. Ma R, Zhang J, Liu X, Li L, Liu H, Rui R, *et al.* Involvement of Rab6a in organelle rearrangement and cytoskeletal organization during mouse oocyte maturation. *Sci. Rep.* 2016, **6**(1): 23560.
5. FitzHarris G, Marangos P, Carroll J. Changes in endoplasmic reticulum structure during mouse oocyte maturation are controlled by the cytoskeleton and cytoplasmic dynein. *Dev Biol.* 2007, **305**(1): 133-144.
6. Cheeseman LP, Boulanger J, Bond LM, Schuh M. Two pathways regulate cortical granule translocation to prevent polyspermy in mouse oocytes. *Nat. Commun.* 2016, **7**: 13726.
7. Xiong B, Zhao YG, Beall S, Sadusky AB, Dean J. A Unique Egg Cortical Granule Localization Motif Is Required for Ovastacin Sequestration to Prevent Premature ZP2 Cleavage and Ensure Female Fertility in Mice. *Plos Genetics* 2017, **13**(1): e1006580.
8. El-Hayek S, Yang Q, Abbassi L, FitzHarris G, Clarke HJ. Mammalian Oocytes Locally Remodel Follicular Architecture to Provide the Foundation for Germline-Soma Communication. *Curr. Biol.* 2018, **28**(7): 1124-1131 e1123.

Reviewers' Comments:

Reviewer #1:

Remarks to the Author:

Dear authors,

Thank you for your responses. The manuscript has improved since its first format.

Reviewer #2:

Remarks to the Author:

The authors have made a substantial and serious effort to address the concerns raised in the previous reviews. Although the interpretation that the phenotype of the knockout is due to a lack of secreted GDF9 remains speculative, there is a lot of new and interesting data here that extends our knowledge of female germ-cell development and will be greatly appreciated by the community.

Rebuttal letter:

Reviewer #1 (Remarks to the Author):

Dear authors,

Thank you for your responses. The manuscript has improved since its first format.

Author response: Thanks for the kindly help. We really appreciated your valuable suggestions to help us improving our study.

Reviewer #2 (Remarks to the Author):

The authors have made a substantial and serious effort to address the concerns raised in the previous reviews. Although the interpretation that the phenotype of the knockout is due to a lack of secreted GDF9 remains speculative, there is a lot of new and interesting data here that extends our knowledge of female germ-cell development and will be greatly appreciated by the community.

Author response: Thanks for the kindly help and suggestion. We have added the discussion that “Also, inconsistency of the phenotypes between *Gdf9*^{-/-} and *Oo-Rdx*^{-/-} females implied complex underlying mechanisms of either Oo-Mvi forming regulations or the OSF release controlling on oocytes for future study.” in the manuscript in line 329-330. Hoping the further study will uncover this point.